# Split LBI: An Iterative Regularization Path with Structural Sparsity

**Chendi Huang**[1], **Xinwei Sun**[1], **Jiechao Xiong**[1], **Yuan Yao**[2,1]
[1]Peking University, [2]Hong Kong University of Science and Technology
{cdhuang, sxwxiaoxiaohehe, xiongjiechao}@pku.edu.cn, yuany@ust.hk

## Abstract

An iterative regularization path with structural sparsity is proposed in this paper based on variable splitting and the Linearized Bregman Iteration, hence called *Split LBI*. Despite its simplicity, Split LBI outperforms the popular generalized Lasso in both theory and experiments. A theory of path consistency is presented that equipped with a proper early stopping, Split LBI may achieve model selection consistency under a family of Irrepresentable Conditions which can be weaker than the necessary and sufficient condition for generalized Lasso. Furthermore, some $\ell_2$ error bounds are also given at the minimax optimal rates. The utility and benefit of the algorithm are illustrated by applications on both traditional image denoising and a novel example on partial order ranking.

## 1 Introduction

In this paper, consider the recovery from linear noisy measurements of $\beta^\star \in \mathbb{R}^p$, which satisfies the following structural sparsity that the linear transformation $\gamma^\star := D\beta^\star$ for some $D \in \mathbb{R}^{m \times p}$ has most of its elements being zeros. For a design matrix $X \in \mathbb{R}^{n \times p}$, let

$$y = X\beta^\star + \epsilon, \ \gamma^\star = D\beta^\star \ (S = \operatorname{supp}(\gamma^\star), \ s = |S|), \tag{1.1}$$

where $\epsilon \in \mathbb{R}^n$ has independent identically distributed components, each of which has a sub-Gaussian distribution with parameter $\sigma^2$ ($\mathbb{E}[\exp(t\epsilon_i)] \leq \exp(\sigma^2 t^2/2)$). Here $\gamma^\star$ is *sparse*, i.e. $s \ll m$. Given $(y, X, D)$, the purpose is to estimate $\beta^\star$ as well as $\gamma^\star$, and in particular, recovers the support of $\gamma^\star$.

There is a large literature on this problem. Perhaps the most popular approach is the following $\ell_1$-penalized convex optimization problem,

$$\arg \min_\beta \left( \frac{1}{2n} \|y - X\beta\|_2^2 + \lambda \|D\beta\|_1 \right). \tag{1.2}$$

Such a problem can be at least traced back to [ROF92] as a *total variation regularization* for image denoising in applied mathematics; in statistics it is formally proposed by [Tib+05] as *fused Lasso*. As $D = I$ it reduces to the well-known *Lasso* [Tib96] and different choices of $D$ include many special cases, it is often called *generalized Lasso* [TT11] in statistics.

Various algorithms are studied for solving (1.2) at fixed values of the tuning parameter $\lambda$, most of which is based on the Split Bregman or ADMM using operator splitting ideas (see for examples [GO09; YX11; Wah+12; RT14; Zhu15] and references therein). To avoid the difficulty in dealing with the structural sparsity in $\|D\beta\|_1$, these algorithms exploit an augmented variable $\gamma$ to enforce sparsity while keeping it close to $D\beta$.

On the other hand, regularization paths are crucial for model selection by computing estimators as functions of regularization parameters. For example, [Efr+04] studies the regularization path of standard Lasso with $D = I$, the algorithm in [Hoe10] computes the regularization path of fused

Lasso, and the dual path algorithm in [TT11] can deal with generalized Lasso. Recently, [AT16] discussed various efficient implementations of the the algorithm in [TT11], and the related R package `genlasso` can be found in CRAN repository. All of these are based on homotopy method of solving convex optimization (1.2).

Our departure here, instead of solving (1.2), is to look at an extremely simple yet novel iterative scheme which finds a new regularization path with structural sparsity. We are going to show that it works in a better way than `genlasso`, in both theory and experiments. To see this, define a loss function which splits $D\beta$ and $\gamma$,

$$\ell(\beta, \gamma) := \frac{1}{2n} \|y - X\beta\|_2^2 + \frac{1}{2\nu} \|\gamma - D\beta\|_2^2 \quad (\nu > 0). \tag{1.3}$$

Now consider the following iterative algorithm,

$$\beta_{k+1} = \beta_k - \kappa\alpha\nabla_\beta\ell(\beta_k, \gamma_k), \tag{1.4a}$$

$$z_{k+1} = z_k - \alpha\nabla_\gamma\ell(\beta_k, \gamma_k), \tag{1.4b}$$

$$\gamma_{k+1} = \kappa \cdot \text{prox}_{\|\cdot\|_1}(z_{k+1}), \tag{1.4c}$$

where the initial choice $z_0 = \gamma_0 = 0 \in \mathbb{R}^m$, $\beta_0 = 0 \in \mathbb{R}^p$, parameters $\kappa > 0$, $\alpha > 0$, $\nu > 0$, and the proximal map associated with a convex function $h$ is defined by $\text{prox}_h(z) = \arg\min_x \|z - x\|^2/2 + h(x)$, which is reduced to the *shrinkage* operator when $h$ is taken to be the $\ell_1$-norm, $\text{prox}_{\|\cdot\|_1}(z) = \mathcal{S}(z, 1)$ where

$$\mathcal{S}(z, \lambda) = \text{sign}(z) \cdot \max(|z| - \lambda, 0) \quad (\lambda \geq 0).$$

In fact, without the sparsity enforcement (1.4c), the algorithm is called the *Landweber Iteration* in inverse problems [YRC07], also known as $L_2$-*Boost* [BY02] in statistics. When $D = I$ and $\nu \to 0$ which enforces $\gamma = D\beta = \beta$, the iteration (1.4) is reduced (by dropping (1.4a)) to the popular *Linearized Bregman Iteration* (LBI) for linear regression or compressed sensing which is firstly proposed in [Yin+08]. The simple iterative scheme returns the whole regularization path, at the same cost of computing one Lasso estimator at a fixed regularization parameter using the iterative soft-thresholding algorithm. However, LBI regularization path could be better than Lasso regularization path which is always biased. In fact, recently [Osh+16] shows that under nearly the same conditions as standard Lasso, LBI may achieve sign-consistency but with a less biased estimator than Lasso, which in the limit dynamics will reach the *bias-free Oracle* estimator.

The difference between (1.4) and the standard LBI lies in the partial sparsity control on $\gamma$, which splits the structural sparsity on $D\beta$ into a sparse $\gamma$ and $D\beta$ by controlling their gap $\|\gamma - D\beta\|^2/(2\nu)$. Thereafter algorithm (1.4) is called *Split LBI* in this paper.

Split LBI generates a sequence $(\beta_k, \gamma_k)_{k \in \mathbb{N}}$ which indeed defines a discrete regularization path. Furthermore, the path can be more accurate than that of generalized Lasso, in terms of *Area Under Curve* (AUC) measurement of the order of regularization paths becoming nonzero in consistent with the ground truth sparsity pattern. The following simple experiment illustrates these properties.

**Example 1.** Consider two problems: standard Lasso and 1-D fused Lasso. In both cases, set $n = p = 50$, and generate $X \in \mathbb{R}^{n \times p}$ denoting $n$ i.i.d. samples from $N(0, I_p)$, $\epsilon \sim N(0, I_n)$, $y = X\beta^\star + \epsilon$. $\beta_j^\star = 2$ (if $1 \leq j \leq 10$), $-2$ (if $11 \leq j \leq 15$), and $0$ (otherwise). For Lasso we choose $D = I$, and for 1-D fused Lasso we choose $D = [D_1; D_2] \in \mathbb{R}^{(p-1+p)\times p}$ such that $(D_1\beta)_j = \beta_j - \beta_{j+1}$ (for $1 \leq j \leq p-1$) and $D_2 = I_p$. The left panel of Figure 1 shows the regularization paths by `genlasso` ($\{D\beta_\lambda\}$) and by iteration (1.4) (linear interpolation of $\{\gamma_k\}$) with $\kappa = 200$ and $\nu \in \{1, 5, 10\}$, respectively. The generalized Lasso path is in fact piecewise linear with respect to $\lambda$ while we show it along $t = 1/\lambda$ for a comparison. Note that the iterative paths exhibit a variety of different shapes depending on the choice of $\nu$. However, in terms of order of those curves entering into nonzero range, these iterative paths exhibit a *better* accuracy than `genlasso`. Table 1 shows this by the mean AUC of 100 independent experiments in each case, where the increase of $\nu$ improves the model selection accuracy of Split LBI paths and beats that of generalized Lasso.

*Why does the simple iterative algorithm* (1.4) *work, even better than the generalized Lasso?* In this paper, we aim to answer it by presenting a theory for model selection consistency of (1.4).

Model selection and estimation consistency of generalized Lasso (1.2) has been studied in previous work. [SSR12] considered the model selection consistency of the edge Lasso, with a special $D$ in

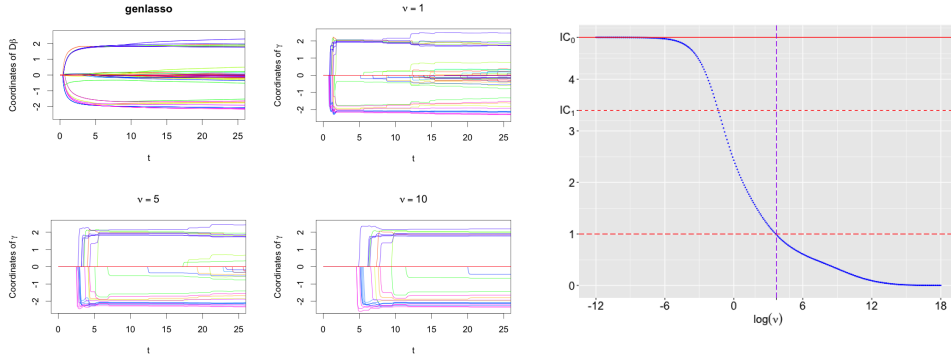

Figure 1: Left shows $\{D\beta_\lambda\}$ $(t = 1/\lambda)$ by `genlasso` and $\{\gamma_k\}$ $(t = k\alpha)$ by Split LBI (1.4) with $\nu = 1, 5, 10$, for 1-D fused Lasso. Right is a comparison between our family of Irrepresentable Condition ($\mathrm{IRR}(\nu)$) and IC in [Vai+13], with log-scale horizontal axis. As $\nu$ grows, $\mathrm{IRR}(\nu)$ can be significantly smaller than $\mathrm{IC}_0$ and $\mathrm{IC}_1$, so that our model selection condition is easier to be met!

Table 1: Mean AUC (with standard deviation) comparisons where Split LBI (1.4) beats `genlasso`. Left is for the standard Lasso. Right is for the 1-D fused Lasso in Example 1.

| genlasso | Split LBI | | | genlasso | Split LBI | | |
|---|---|---|---|---|---|---|---|
| | 1 | 5 | 10 | | 1 | 5 | 10 |
| .9426 | .9845 | .9969 | **.9982** | .9705 | .9955 | .9996 | **.9998** |
| (.0390) | (.0185) | (.0065) | (**.0043**) | (.0212) | (.0056) | (.0014) | (**.0009**) |

(1.2), which has applications over graphs. [LYY13] provides an upper bound of estimation error by assuming the design matrix $X$ is a Gaussian random matrix. In particular, [Vai+13] proposes a general condition called *Identifiability Criterion* (IC) for sign consistency. [LST13] establishes a general framework for model selection consistency for penalized M-estimators, proposing an Irrepresentable Condition which is equivalent to IC from [Vai+13] under the specific setting of (1.2). In fact both of these conditions are sufficient and necessary for structural sparse recovery by generalized Lasso (1.2) in a certain sense.

However, as we shall see soon, the benefits of exploiting algorithm (1.4) not only lie in its algorithmic simplicity, but also provide a possibility of theoretical improvement on model selection consistency. Below a new family of Irrepresentable Condition depending on $\nu$ will be presented for iteration (1.4), under which model selection consistency can be established. Moreover, this family can be *weaker* than IC as the parameter $\nu$ grows, which sheds light on the superb performance of Split LBI we observed above. The main contributions of this paper can be summarized as follows: (A) a new iterative regularization path with structural sparsity by (1.4); (B) a theory of path consistency which shows the model selection consistency of (1.4), under some weaker conditions than generalized Lasso, together with $\ell_2$ error bounds at minimax optimal rates. Further experiments are given with applications on 2-D image reconstruction and partial order estimation.

## 1.1 Notation

For matrix $Q$ with $m$ rows ($D$ for example) and $J \subseteq \{1, 2, \ldots, m\}$, let $Q_J = Q_{J,\cdot}$ be the submatrix of $Q$ with rows indexed by $J$. However, for $Q \in \mathbb{R}^{n \times p}$ ($X$ for example) and $J \subseteq \{1, 2, \ldots, p\}$, let $Q_J = Q_{\cdot, J}$ be the submatrix of $Q$ with columns indexed by $J$, abusing the notation.

Sometimes we use $\langle a, b \rangle := a^T b$, denoting the inner product between vectors $a, b$. $P_L$ denotes the projection matrix onto a linear subspace $L$, Let $L_1 + L_2 := \{\xi_1 + \xi_2 : \xi \in L_1, \xi \in L_2\}$ for subspaces $L_1, L_2$. For a matrix $Q$, let $Q^\dagger$ denotes the Moore-Penrose pseudoinverse of $Q$, and we recall that $Q^\dagger = (Q^T Q)^\dagger Q^T$. Let $\lambda_{\min}(Q), \lambda_{\max}(Q)$ denotes the smallest and largest singular value (i.e. eigenvalue if $Q$ is symmetric) of $Q$. For symmetric matrices $P$ and $Q$, $Q \succ P$ (or $Q \succeq P$) means that $Q - P$ is positive (semi)-definite, respectively. Let $Q^* := Q^T/n$.

## 2 Path Consistency of Split LBI

### 2.1 Basic Assumptions

For the identifiability of $\beta^\star$, we assume that $\beta^\star$ and its estimators of interest are restricted in

$$L := (\ker(X) \cap \ker(D))^\perp = \operatorname{Im}\left(X^T\right) + \operatorname{Im}\left(D^T\right),$$

since replacing $\beta^\star$ with "the projection of $\beta^\star$ onto $L$" does not change the model.

Note that $\ell(\beta, \gamma)$ is quadratic, and we can define its Hessian matrix which depends on $\nu > 0$

$$H(\nu) := \nabla^2 \ell(\beta, \gamma) \equiv \begin{pmatrix} X^*X + D^TD/\nu & -D^T/\nu \\ -D/\nu & I_m/\nu \end{pmatrix}. \tag{2.1}$$

We make the following assumptions on $H$.

**Assumption 1** (Restricted Strong Convexity (RSC)). There is a constant $\lambda_H > 0$ such that

$$\left(\beta^T, \gamma_S^T\right) \cdot H_{(\beta,S),(\beta,S)} \cdot \begin{pmatrix} \beta \\ \gamma_S \end{pmatrix} \geq \lambda_H \left\| \begin{pmatrix} \beta \\ \gamma_S \end{pmatrix} \right\|_2^2 \; (\beta \in L, \; \gamma_S \in \mathbb{R}^s). \tag{2.2}$$

*Remark* 1. Since the true parameter $\operatorname{supp}(\gamma^\star) = \operatorname{supp}(D\beta^\star) = S$, it is equivalent to say that the loss $\ell(\beta, \gamma)$ is strongly convex when restricting on the sparse subspace corresponding to support of $\gamma^\star$.

**Assumption 2** (Irrepresentable Condition (IRR)). There is a constant $\eta \in (0, 1]$ such that

$$\sup_{\rho \in [-1,1]^s} \left\| H_{S^c,(\beta,S)} H_{(\beta,S),(\beta,S)}^\dagger \cdot \begin{pmatrix} 0_p \\ \rho \end{pmatrix} \right\|_\infty \leq 1 - \eta. \tag{2.3}$$

*Remark* 2. IRR here directly generalizes the Irrepresentable Condition from standard Lasso [ZY06] and other algorithms [Tro04], to the partial Lasso: $\min_{\beta,\gamma}(\ell(\beta, \gamma) + \lambda\|\gamma\|_1)$. Following the standard Lasso, one version of the Irrepresentable Condition should be

$$\left\| H_{S^c,(\beta,S)} H_{(\beta,S),(\beta,S)}^\dagger \rho_{(\beta,S)}^\star \right\|_\infty \leq 1 - \eta, \text{ where } \rho_{(\beta,S)}^\star = \begin{pmatrix} 0_p \\ \rho_S^\star \end{pmatrix}.$$

$\rho_{(\beta,S)}^\star$ is the value of gradient (subgradient) of $\ell_1$ penalty function $\|\cdot\|_1$ on $(\beta^\star; \gamma_S^\star)$. Here $\rho_\beta^\star = 0_p$, because $\beta$ is not assumed to be sparse and hence is not penalized. Assumption 2 slightly strengthens this by a supremum over $\rho$, for uniform sparse recovery independent to a particular sign pattern of $\gamma^\star$.

### 2.2 Equivalent Conditions and a Comparison Theorem

The assumptions above, though being natural, are not convenient to compare with that in [Vai+13]. Here we present some equivalent conditions, followed by a comparison theorem showing that IRR can be weaker than IC in [Vai+13], a necessary and sufficient for model selection consistency of generalized Lasso.

First of all, we introduce some notations. Given $\gamma$, minimizing $\ell$ solves $\beta = A^\dagger(\nu X^*y + D^T\gamma)$, where $A := \nu X^*X + D^TD$. Substituting $A^\dagger(\nu X^*y + D^T\gamma_k)$ for $\beta_k$ in (1.4b), and dropping (1.4a), we have

$$z_{k+1} = z_k + \alpha(DA^\dagger X^*y - \Sigma\gamma_k), \tag{2.4a}$$

$$\gamma_{k+1} = \kappa \cdot \operatorname{prox}_{\|\cdot\|_1}(z_{k+1}), \tag{2.4b}$$

where

$$\Sigma := \left(I - DA^\dagger D^T\right)/\nu, \; A = \nu X^*X + D^TD. \tag{2.5}$$

In other words, $\Sigma$ is the Schur complement of $H_{\beta,\beta}$ in Hessian matrix $H(\nu)$. Comparing (2.4) with the standard LBI ($D = I$) studied in [Osh+16], we know that $\Sigma$ in our paper plays the similar role of $X^*X$ in their paper. In order to obtain path consistency results of standard LBI in [Osh+16], they propose "Restricted Strong Convexity" and "Irrpresentable Condition" on $X^*X$. So in this paper, we can obtain similar assumptions on $\Sigma$ (instead of $H$), which actually prove to be equivalent with Assumption 1 and 2, and closely related to literature.

Precisely, by Lemma 6 in Supplementary Information we know that Assumption 1 is equivalent to

**Assumption 1′** (Restricted Strong convexity (RSC)). There is a constant $\lambda_\Sigma > 0$ such that

$$\Sigma_{S,S} \succeq \lambda_\Sigma I. \tag{2.6}$$

*Remark* 3. Lemma 2 in Supplementary Information says $\Sigma_{S,S} \succ 0 \Leftrightarrow \ker(D_{S^c}) \cap \ker(X) \subseteq \ker(D_S)$, which is also a natural assumption for the uniqueness of $\beta^\star$. Actually, if it fails, then there will be some $\beta$ such that $D_{S^c}\beta = 0$, $X\beta = 0$ while $D_S\beta \neq 0$. Thus for any $\beta'^\star := \beta^\star + \beta$, we have $y = X\beta'^\star + \epsilon$, $\mathrm{supp}(D\beta'^\star) \subseteq \mathrm{supp}(D\beta^\star) = S$, while $D_S\beta'^\star \neq D_S\beta^\star$. Therefore one can neither estimate $\beta^\star$ nor $D_S\beta^\star$ even if the support set is known or has been exactly recovered.

When $\Sigma_{S,S} \succ 0$, Lemma 7 in Supplementary Information implies that Assumption 2 is equivalent to

**Assumption 2′** (Irrepresentable condition (IRR)). There is a constant $\eta \in (0, 1]$ such that

$$\left\| \Sigma_{S^c,S} \Sigma_{S,S}^{-1} \right\|_\infty \leq 1 - \eta. \tag{2.7}$$

*Remark* 4. For standard Lasso problems ($D = I$), it is easy to derive $\Sigma = X^*(1 + \nu XX^*)^{-1}X \approx X^*X$ when $\nu$ is small. So Assumption 1′ approximates the usual Restricted Strong Convexity assumption $X_S^*X_S \succeq \lambda_\Sigma I$ and Assumption 2′ approximates the usual Irrepresentable Condition $\|X_{S^c}^*X_S(X_S^*X_S)^{-1}\|_\infty \leq 1 - \eta$ for standard Lasso problems.

The left hand side of (2.7) depends on parameter $\nu$. From now on, define

$$\mathrm{IRR}(\nu) := \left\| \Sigma_{S^c,S} \Sigma_{S,S}^{-1} \right\|_\infty, \ \mathrm{IRR}(0) := \lim_{\nu \to 0} \mathrm{IRR}(\nu), \ \mathrm{IRR}(\infty) := \lim_{\nu \to +\infty} \mathrm{IRR}(\nu). \tag{2.8}$$

Now we are going to compare Assumption 2′ with the assumption in [Vai+13]. Let $W$ be a matrix whose columns form an orthogonal basis of $\ker(D_{S^c})$, and define

$$\Omega^S := \left(D_{S^c}^\dagger\right)^T \left(X^*XW \left(W^TX^*XW\right)^\dagger W^T - I\right) D_S^T,$$

$$\mathrm{IC}_0 := \left\|\Omega^S\right\|_\infty, \ \mathrm{IC}_1 := \min_{u \in \ker\left(D_{S^c}^T\right)} \left\|\Omega^S \mathrm{sign}\left(D_S\beta^\star\right) - u\right\|_\infty.$$

[Vai+13] proved the sign consistency of the generalized Lasso estimator of (1.2) for specifically chosen $\lambda$, under the assumption $\mathrm{IC}_1 < 1$ along with $\ker(D_{S^c}) \cap \ker(X) = \{0\}$. As we shall see later, the same conclusion holds under the assumption $\mathrm{IRR}(\nu) \leq 1 - \eta$ along with Assumption 1′ which is equivalent to $\ker(D_{S^c}) \cap \ker(X) \subseteq \ker(D_S)$. *Which assumption is weaker to be satisfied?* The following theorem answers this, whose proof is in Supplementary Information.

**Theorem 1** (Comparisons between IRR in Assumption 2′ and IC in [Vai+13]).

  *1.* $\mathrm{IC}_0 \geq \mathrm{IC}_1$.

  *2.* $\mathrm{IRR}(0)$ *exists, and* $\mathrm{IRR}(0) = \mathrm{IC}_0$.

  *3.* $\mathrm{IRR}(\infty)$ *exists, and* $\mathrm{IRR}(\infty) = 0$ *if and only if* $\ker(X) \subseteq \ker(D_S)$.

From this comparison theorem with a design matrix $X$ of full column rank, as $\nu$ grows, $\mathrm{IRR}(\nu) < \mathrm{IC}_1 \leq \mathrm{IC}_0$, hence Assumption 2′ is weaker than IC. Now recall the setting of Example 1 where $\ker(X) = 0$ generically. In the right panel of Figure 1, the (solid and dashed) horizontal red lines denote $\mathrm{IC}_0, \mathrm{IC}_1$, and we see the blue curve denoting $\mathrm{IRR}(\nu)$ approaches $\mathrm{IC}_0$ when $\nu \to 0$ and approaches 0 when $\nu \to +\infty$, which illustrates Theorem 1 (here each of $\mathrm{IC}_0, \mathrm{IC}_1, \mathrm{IRR}(\nu)$ is the mean of 100 values calculated under 100 generated $X$'s). Although $\mathrm{IRR}(0) = \mathrm{IC}_0$ is slightly larger than $\mathrm{IC}_1$, $\mathrm{IRR}(\nu)$ can be significantly smaller than $\mathrm{IC}_1$ if $\nu$ is not tiny. On the right side of the vertical line, $\mathrm{IRR}(\nu)$ drops below 1, indicating that Assumption 2′ is satisfied while the assumption in [Vai+13] fails.

*Remark* 5. Despite that Theorem 1 suggests to adopt a large $\nu$, $\nu$ can not be arbitrarily large. From Assumption 1′ and the definition of $\Sigma$, $1/\nu \geq \|\Sigma\|_2 \geq \|\Sigma_{S,S}\|_2 \geq \lambda_\Sigma$. So if $\nu$ is too large, $\lambda_\Sigma$ has to be small enough, which will deteriorates the estimator in terms of $\ell_2$ error shown in the next.

## 2.3 Consistency of Split LBI

We are ready to establish the theorems on path consistency of Split LBI (1.4), under Assumption 1 and 2. The proofs are based on a careful treatment of the limit dynamics of (1.4) and collected in Supplementary Information. Before stating the theorems, we need some definitions and constants.

Let the compact singular value decomposition (compact SVD) of $D$ be

$$D = U\Lambda V^T \; \left(\Lambda \in \mathbb{R}^{r \times r}, \; \Lambda \succ 0, \; U \in \mathbb{R}^{m \times r}, \; V \in \mathbb{R}^{p \times r}\right), \qquad (2.9)$$

and $(V, \tilde{V})$ be an orthogonal square matrix. Let the compact SVD of $X\tilde{V}/\sqrt{n}$ be

$$X\tilde{V}/\sqrt{n} = U_1 \Lambda_1 V_1^T \; \left(\Lambda_1 \in \mathbb{R}^{r' \times r'}, \; \Lambda_1 \succ 0, \; U_1 \in \mathbb{R}^{n \times r'}, \; V_1 \in \mathbb{R}^{(p-r) \times r'}\right), \qquad (2.10)$$

and let $(V_1, \tilde{V}_1)$ be an orthogonal square matrix. Let

$$\Lambda_X = \sqrt{\Lambda_{\max}\left(X^*X\right)}, \; \lambda_D = \lambda_{\min}\left(\Lambda\right), \; \Lambda_D = \Lambda_{\max}\left(\Lambda\right), \; \lambda_1 = \lambda_{\min}\left(\Lambda_1\right). \qquad (2.11)$$

We see $\Lambda_D$ is the largest singular value of $D$, $\lambda_D$ is the smallest *nonzero* singular value of $D$, and $\lambda_1^2$ is the smallest *nonzero* eigenvalue of $\tilde{V}^T X^* X \tilde{V}$. If $D$ has full column rank, then $r = p$, $r' = 0$, and $\tilde{V}, U_1, \Lambda_1, V_1, \lambda_1$ all drop, while $\tilde{V}_1 \in \mathbb{R}^{(p-r) \times (p-r)}$ is an orthogonal square matrix.

The following theorem says that under Assumption 1 and 2, Split LBI will automatically evolve in an "*oracle*" subspace (unknown to us) restricted within the support set of $(\beta^\star, \gamma^\star)$ before leaving it, and if the signal parameters is strong enough, sign consistency will be reached. Moreover, $\ell_2$ error bounds on $\gamma_k$ and $\beta_k$ are given.

**Theorem 2** (Consistency of Split LBI). *Under Assumption 1 and 2, suppose $\kappa$ is large enough to satisfy*

$$\kappa \geq \frac{4}{\eta}\left(1 + \frac{1}{\lambda_D} + \frac{\Lambda_X}{\lambda_1 \lambda_D}\right)\left(1 + \sqrt{\frac{2\left(1 + \nu\Lambda_X^2 + \Lambda_D^2\right)}{\lambda_H \nu}}\right)$$

$$\cdot \left((1 + \Lambda_D)\|\beta^\star\|_2 + \frac{2\sigma}{\lambda_H}\left(\frac{\Lambda_X}{\lambda_D} + \frac{\Lambda_X}{\lambda_D^2} + \frac{\lambda_H \lambda_D^2 + \Lambda_X^2}{\lambda_1 \lambda_D^2}\right)\right), \quad (2.12)$$

*and $\kappa\alpha\|H\|_2 < 2$. Let*

$$\bar{\tau} := \frac{\eta}{8\sigma} \cdot \frac{\lambda_D}{\Lambda_X}\sqrt{\frac{n}{\log m}}, \; K := \left\lfloor \frac{\bar{\tau}}{\alpha} \right\rfloor, \; \lambda_H' := \lambda_H(1 - \kappa\alpha\|H\|_2/2) > 0.$$

*Then with probability not less than $1 - 6/m - 3\exp(-4n/5)$, we have all the following properties.*

1. No-false-positive*: The solution has no false-positive, i.e. $\operatorname{supp}(\gamma_k) \subseteq S$, for $0 \leq k\alpha \leq \bar{\tau}$.*

2. Sign consistency of $\gamma_k$*: Once the signal is strong enough such that*

$$\gamma_{\min}^\star := (D_S\beta^\star)_{\min} \geq \frac{16\sigma}{\eta\lambda_H'\left(1 - 5\alpha/\bar{\tau}\right)} \cdot \frac{\Lambda_X\Lambda_D}{\lambda_D^2}\left(2\log s + 5 + \log(8\Lambda_D)\right)\sqrt{\frac{\log m}{n}},$$

$$(2.13)$$

*then $\gamma_k$ has sign consistency at $K$, i.e. $\operatorname{sign}\left(\gamma_K\right) = \operatorname{sign}\left(D\beta^\star\right)$.*

3. $\ell_2$ consistency of $\gamma_k$*:*

$$\|\gamma_K - D\beta^\star\|_2 \leq \frac{42\sigma}{\eta\lambda_H'\left(1 - \alpha/\bar{\tau}\right)} \cdot \frac{\Lambda_X}{\lambda_D}\sqrt{\frac{s\log m}{n}}.$$

4. $\ell_2$ consistency of $\beta_k$*:*

$$\|\beta_K - \beta^\star\|_2 \leq \frac{42\sigma}{\eta\lambda_H'\left(1 - \alpha/\bar{\tau}\right)} \cdot \frac{\lambda_1\Lambda_X(1 + \lambda_D) + \Lambda_X^2}{\lambda_1\lambda_D^2}\sqrt{\frac{s\log m}{n}} + \frac{2\sigma}{\lambda_1}\sqrt{\frac{r'\log m}{n}}$$

$$+ \nu \cdot 2\sigma \cdot \frac{\lambda_1\Lambda_X + \Lambda_X^2}{\lambda_1\lambda_D^2}.$$

Despite that the sign consistency of $\gamma_k$ can be established here, usually one can not expect $D\beta_k$ recovers the sparsity pattern of $\gamma^\star$ due to the variable splitting. As shown in the last term of $\ell_2$ error bound of $\beta_k$, increasing $\nu$ will sacrifice its accuracy. However, one can remedy this by projecting $\beta_k$ on to a subspace using the support set of $\gamma_k$, and obtain a good estimator $\tilde{\beta}_k$ with both sign consistency and $\ell_2$ consistency at the minimax optimal rates.

**Theorem 3** (Consistency of revised version of Split LBI). *Under Assumption 1 and 2, suppose $\kappa$ is large enough to satisfy (2.12), and $\kappa\alpha\|H\|_2 < 2$. $\bar{\tau}, K, \lambda'_H$ are defined the same as in Theorem 2. Define*

$$S_k := \text{supp}(\gamma_k), \ P_{S_k} := P_{\ker\left(D_{S_k^c}\right)} = I - D_{S_k^c}^\dagger D_{S_k^c}, \ \tilde{\beta}_k := P_{S_k}\beta_k.$$

*If $S_k^c = \varnothing$, define $P_{S_k} = I$. Then we have the following properties.*

1. Sign consistency of $\tilde{\beta}_k$: *If the $\gamma^\star_{\min}$ condition (2.13) holds, then with probability not less than $1 - 8/m - 3\exp(-4n/5)$, there holds $\text{sign}(D\tilde{\beta}_K) = \text{sign}(D\beta^\star)$.*

2. $\ell_2$ consistency of $\tilde{\beta}_k$: *With probability not less than $1 - 8/m - 2r'/m^2 - 3\exp(-4n/5)$, we have that for $0 \leq k\alpha \leq \bar{\tau}$,*

$$\left\|\tilde{\beta}_k - \beta^\star\right\|_2 \leq \left(\frac{10\sqrt{s}}{\lambda'_H k\alpha} + \frac{2\sigma}{\lambda'_H} \cdot \frac{\Lambda_X\Lambda_D}{\lambda_D^3}\sqrt{\frac{s\log m}{n}}\right)$$
$$+ \frac{2\sigma}{\lambda'_H}\left(\frac{\Lambda_X}{\lambda_D^2} + \frac{\lambda'_H\lambda_D^2 + \Lambda_X^2}{\lambda_1\lambda_D^2}\right)\sqrt{\frac{r'\log m}{n}} + 2\left\|D_{S_k^c}^\dagger D_{S_k^c \cap S}\beta^\star\right\|_2.$$

*Consequently, if additionally $S_K = S$, then the last term on the right hand side drops for $k = K$, and it reaches*

$$\left\|\tilde{\beta}_K - \beta^\star\right\|_2 \leq \frac{80\sigma}{\eta\lambda'_H(1 - \alpha/\bar{\tau})} \cdot \frac{\Lambda_X\left(\Lambda_D + \lambda_D^2\right)}{\lambda_D^3}\sqrt{\frac{s\log m}{n}}$$
$$+ \frac{2\sigma}{\lambda'_H}\left(\frac{\Lambda_X}{\lambda_D^2} + \frac{\lambda'_H\lambda_D^2 + \Lambda_X^2}{\lambda_1\lambda_D^2}\right)\sqrt{\frac{r'\log m}{n}}.$$

*Remark* 6. Note that $r' \leq \min(n, p-r)$. In many real applications, $r'$ is very small. So the dominant $\ell_2$ error rate is $O(\sqrt{s\log m/n})$, which is minimax optimal [LST13; LYY13].

## 3 Experiments

### 3.1 Parameter Setting

Parameter $\kappa$ should be large enough according to (2.12). Moreover, step size $\alpha$ should be small enough to ensure the stability of Split LBI. When $\nu, \kappa$ are determined, $\alpha$ can actually be determined by $\alpha = \nu/(\kappa(1 + \nu\Lambda_X^2 + \Lambda_D^2))$ (see (C.6) in Supplementary Information).

### 3.2 Application: Image Denoising

Consider the image denoising problem in [TT11]. The original image is resized to $50 \times 50$, and reset with only four colors, as in the top left image in Figure 2. Some noise is added by randomly changing some pixels to be white, as in the bottom left. Let $G = (V, E)$ is the 4-nearest-neighbor grid graph on pixels, then $\beta = (\beta_R, \beta_G, \beta_B) \in \mathbb{R}^{3|V|}$ since there are 3 color channels (RGB channels). $X = I_{3|V|}$ and $D = \text{diag}(D_G, D_G, D_G)$, where $D_G\delta \in \mathbb{R}^{|E|\times|V|}$ is the gradient operator on graph $G$ defined by $(D_Gx)(e_{ij}) = x_i - x_j, e_{ij} \in E$. Set $\nu = 180$, $\kappa = 100$. The regularization path of Split LBI is shown in Figure 2, where as $t$ evolves, images on the path gradually select visually salient features before picking up the random noise. Now compare the AUC (Area Under Curve) of `genlasso` and Split LBI algorithm with different $\nu$. For simplicity we show the AUC corresponding to the red color channel. Here $\nu \in \{1, 20, 40, 60, \ldots, 300\}$. As shown in the right panel of Figure 2, with the increase of $\nu$, Split LBI beats `genlasso` with higher AUC values.

### 3.3 Application: Partial Order Ranking for Basketball Teams

Here we consider a new application on the ranking of $p = 12$ FIBA basketball teams into partial orders. The teams are listed in Figure 3. We collected $n = 134$ pairwise comparison game results mainly from various important championship such as Olympic Games, FIBA World Championship

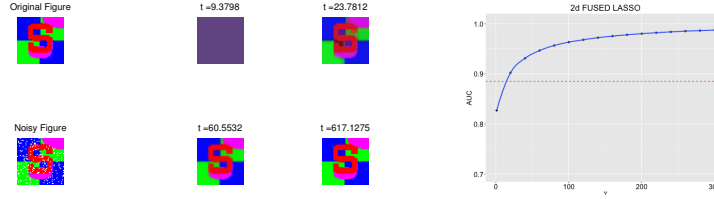

Figure 2: Left is image denoising results by Split LBI. Right shows the AUC of Split LBI (blue solid line) increases and exceeds that of `genlasso` (dashed red line) as $\nu$ increases.

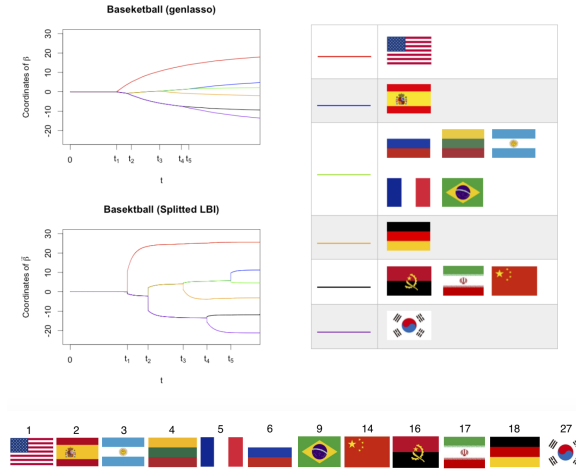

Figure 3: Partial order ranking for basketball teams. Top left: $\{\beta_\lambda\}$ ($t = 1/\lambda$) by `genlasso` and $\tilde{\beta}_k$ ($t = k\alpha$) by Split LBI. Top right: grouping result just passing $t_5$. Bottom: FIBA ranking.

and FIBA Basketball Championship in 5 continents from 2006–2014 (8 years is not too long for teams to keep relatively stable levels while not too short to have enough samples). For each sample indexed by $k$ and corresponding team pair $(i, j)$, $y_k = s_i - s_j$ is the score difference between team $i$ and $j$. We assume a model $y_k = \beta^\star_{i_k} - \beta^\star_{j_k} + \epsilon_k$ where $\beta^\star \in \mathbb{R}^p$ measures the strength of these teams. So the design matrix $X \in \mathbb{R}^{n \times p}$ is defined by its $k$-th row: $x_{k,i_k} = 1$, $x_{k,j_k} = -1$, $x_{k,l} = 0$ ($l \neq i_k, j_k$). In sports, teams of similar strength often meet than those in different levels. Thus we hope to find a coarse grained partial order ranking by adding a structural sparsity on $D\beta^\star$ where $D = cX$ ($c$ scales the smallest nonzero singular value of $D$ to be 1).

The top left panel of Figure 3 shows $\{\beta_\lambda\}$ by `genlasso` and $\tilde{\beta}_k$ by Split LBI with $\nu = 1$ and $\kappa = 100$. Both paths give the same partial order at early stages, though the Split LBI path looks qualitatively better. For example, the top right panel shows the same partial order after the change point $t_5$. It is interesting to compare it against the FIBA ranking in September, 2014, shown in the bottom. Note that the average basketball level in Europe is higher than that of in Asia and Africa, hence China can get more FIBA points than Germany based on the dominant position in Asia, so is Angola in Africa. But their true levels might be lower than Germany, as indicated in our results. Moreover, America (FIBA points 1040.0) itself forms a group, agreeing with the common sense that it is much better than any other country. Spain, having much higher FIBA ranking points (705.0) than the 3rd team Argentina (455.0), also forms a group alone. It is the only team that can challenge America in recent years, and it enters both finals against America in 2008 and 2012.

### Acknowledgments

The authors were supported in part by National Basic Research Program of China under grants 2012CB825501 and 2015CB856000, as well as NSFC grants 61071157 and 11421110001.

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
