[Supplementary Material · lb_split_10-31.pdf]

# Supplementary Information

## A   Some Useful Technical Lemmas

**Lemma 1.** *Suppose that $\epsilon \in \mathbb{R}^n$ has independent identically distributed components, each of which has a sub-Gaussian distribution with parameter $\sigma^2$, then*

$$\mathbb{P}\left(\frac{\|B\epsilon\|_\infty}{\sigma} \geq z\right) \leq 2q \exp\left(-\frac{z^2}{2\|B\|_2^2}\right) \ \left(B \in \mathbb{R}^{q \times n}, \ z \geq 0\right), \tag{A.1}$$

$$\mathbb{P}\left(\frac{\|\epsilon\|_2^2}{n\sigma^2} \geq 1 + z\right) \leq \exp\left(-\frac{n(z - \log(1+z))}{2}\right) \ (z \geq 0). \tag{A.2}$$

*Moreover, by* (A.1) *we have that for $B \in \mathbb{R}^{q \times n}$, with probability not less than $1 - 2q/m^2$,*

$$\|B\epsilon\|_\infty \leq 2\sigma \cdot \|B\|_2 \sqrt{\log m}. \tag{A.3}$$

*By* (A.2) *we have that with probability not less than $1 - \exp(-4n/5)$,*

$$\|\epsilon\|_2 \leq 2\sigma\sqrt{n}. \tag{A.4}$$

*Proof.* As for (A.1), let $B = (B_{i,j})_{q \times n}$ and $1 \leq i \leq q$, it is well-known that
$$B_{i,\cdot}\epsilon = B_{i,1}\epsilon_1 + B_{i,2}\epsilon_2 + \cdots + B_{i,n}\epsilon_n$$
is also sub-Gaussian, with parameter $b_i^2 = (B_{i,1}^2 + \cdots + B_{i,n}^2)\sigma^2$. Thus

$$\mathbb{P}(\|B\epsilon\|_\infty \geq z) \leq q \cdot \max_{1 \leq i \leq q} \mathbb{P}(|B_{i,\cdot}\epsilon| \geq z) \leq 2q \exp\left(-\frac{z^2}{2b_i^2}\right) \leq 2q \exp\left(-\frac{z^2}{2\|B\|_2^2}\right).$$

As for (A.2), note that for $0 \leq \zeta < 1/2$,

$$\mathbb{P}\left(\frac{\|\epsilon\|_2^2}{n\sigma^2} \geq 1 + z\right) \leq \mathbb{P}\left(\exp\left(\frac{\zeta\|\epsilon\|_2^2}{\sigma^2}\right) \geq \exp(\zeta n(1+z))\right)$$

$$\leq \exp(-\zeta n(1+z))\, \mathbb{E}\left[\exp\left(\frac{\zeta\|\epsilon\|_2^2}{\sigma^2}\right)\right] = \exp(-\zeta n(1+z))\left(\mathbb{E}\left[\exp\left(\frac{\zeta\epsilon_1^2}{\sigma^2}\right)\right]\right)^n$$

$$\leq \exp(-\zeta n(1+z)) \cdot \left(\frac{1}{1-2\zeta}\right)^{n/2}.$$

Take $\zeta = z/(2(1+z)) \in [0, 1/2)$, and (A.2) follows. $\qquad\square$

**Lemma 2.** $\Sigma_{S,S} \succ 0$ *if and only if* $\ker(D_{S^c}) \cap \ker(X) \subseteq \ker(D_S)$.

*Proof.* Define $Q = (D_S^T, D_{S^c}^T, \sqrt{\nu}X^T/\sqrt{n})^T$, and note that for any $\xi \in \mathbb{R}^s$,

$$\left\|\begin{pmatrix}\xi\\0\\0\end{pmatrix} - Q(Q^TQ)^\dagger Q^T \begin{pmatrix}\xi\\0\\0\end{pmatrix}\right\|_2^2 = \begin{pmatrix}\xi\\0\\0\end{pmatrix}^T (I_m - Q(Q^TQ)Q^T)\begin{pmatrix}\xi\\0\\0\end{pmatrix} = \xi^T\Sigma_{S,S}\xi. \tag{A.5}$$

If $\ker(D_{S^c}) \cap \ker(X) \subseteq \ker(D_S)$, then for any $\xi \in \mathbb{R}^s$ satisfying $\xi^T\Sigma_{S,S}\xi = 0$, (A.5) leads to $(\xi^T, 0, 0)^T = Q\beta$ for some $\beta$, implying
$$\xi - D_S\beta = 0, \ D_{S^c}\beta = 0, \ X\beta = 0 \implies \beta \in \ker(D_{S^c}) \cap \ker(X) \subseteq \ker(D_S) \implies \xi = D_S\beta = 0.$$
Therefore, $\Sigma_{S,S} \succ 0$. Conversely, if $\Sigma_{S,S} \succ 0$, then for any $\beta \in \ker(D_{S^c}) \cap \ker(X)$, since

$$(D_S\beta)^T \Sigma_{S,S}(D_S\beta) = \left\|\begin{pmatrix}D_S\beta\\0\\0\end{pmatrix} - Q(Q^TQ)^\dagger Q^T\begin{pmatrix}D_S\beta\\0\\0\end{pmatrix}\right\|_2^2 = \min_{\beta'}\left\|\begin{pmatrix}D_S\beta\\0\\0\end{pmatrix} - Q\beta'\right\|_2^2$$

$$\leq \left\|\begin{pmatrix}D_S\beta\\0\\0\end{pmatrix} - Q\beta\right\|_2^2 = \left\|\begin{pmatrix}D_S\beta\\0\\0\end{pmatrix} - \begin{pmatrix}D_S\beta\\D_{S^c}\beta\\\sqrt{\nu}X\beta/\sqrt{n}\end{pmatrix}\right\|_2^2 = 0,$$

which implies $D_S\beta = 0$, i.e. $\beta \in \ker(D_S)$. So $\ker(D_{S^c}) \cap \ker(X) \subseteq \ker(D_S)$. $\qquad\square$

**Lemma 3.** *Adopt the notation from* (2.9) *to* (2.11). $\beta \in L$ *if and only if*

$$\beta = V\delta + \tilde{V}V_1\xi, \text{ where } \delta = V^T\beta, \ \xi = V_1^T\tilde{V}^T\beta.$$

*Proof.* Note that

$$I = VV^T + \tilde{V}\tilde{V}^T = VV^T + \tilde{V}\left(V_1V_1^T + \tilde{V}_1\tilde{V}_1^T\right)\tilde{V}^T.$$

Right multiplying $\beta$ on both side leads to

$$\beta = V\delta + \tilde{V}V_1\xi + \tilde{V}\tilde{V}_1\left(\tilde{V}_1^T\tilde{V}^T\beta\right). \tag{A.6}$$

It suffices to show

$$\ker\left(\tilde{V}_1^T\tilde{V}^T\right) = L, \text{ which is equivalent to } L' := \text{Im}\left(\tilde{V}\tilde{V}_1\right) = L^{\perp}\left(= \ker(X) \cap \ker(D)\right).$$

For any $\beta \in L'$, we have $X\beta = 0$, $D\beta = 0$ since $X\tilde{V}\tilde{V}_1 = 0$, $D\tilde{V} = 0$, so $\beta \in L^{\perp}$. Conversely, if $\beta \in L^{\perp}$, left multiplying $D$ on both sides of (A.6) leads to $\delta = 0$. Then left multiplying $X$ on both sides of (A.6) further leads to $\xi = 0$. Now (A.6) tells that $\beta \in L'$. So $L' = L^{\perp}$. $\square$

**Lemma 4.** *Adopt the notation from* (2.9) *to* (2.11). *Define* $B := \Lambda^2 + \nu V^T X^*\left(I - U_1 U_1^T\right) XV$. *We have*

$$DA^{\dagger} = U\Lambda B^{-1}V^T\left(I - \frac{1}{\sqrt{n}}X^T U_1 \Lambda_1^{-1}V_1^T\tilde{V}^T\right). \tag{A.7}$$

*Consequently,*

$$\Sigma = \left(I - DA^{\dagger}D^T\right)/\nu = \left(I - U\Lambda B^{-1}\Lambda U^T\right)/\nu. \tag{A.8}$$

*Proof.* Note that

$$\begin{pmatrix} V^T \\ \tilde{V}^T \end{pmatrix} A\left(V, \tilde{V}\right) = \begin{pmatrix} \Lambda^2 + \nu V^T X^* XV & \nu V^T X^T U_1 \Lambda_1 V_1^T/\sqrt{n} \\ \nu V_1 \Lambda_1 U_1^T XV/\sqrt{n} & \nu V_1 \Lambda_1^2 V_1^T \end{pmatrix}$$

$$= QMQ^T, \text{ where } Q := \begin{pmatrix} I_r & V^T X^T U_1 \Lambda_1^{-1}V_1^T/\sqrt{n} \\ 0 & I_{p-r} \end{pmatrix}, \ M := \begin{pmatrix} B & 0 \\ 0 & \nu V_1 \Lambda_1^2 V_1^T \end{pmatrix}$$

We can directly verify that $(QMQ^T)^{\dagger} = (Q^T)^{-1}M^{\dagger}Q^{-1}$, thus

$$DA^{\dagger} = D\left(V, \tilde{V}\right)\left(\begin{pmatrix} V^T \\ \tilde{V}^T \end{pmatrix} A\left(V, \tilde{V}\right)\right)^{\dagger}\begin{pmatrix} V^T \\ \tilde{V}^T \end{pmatrix} = (U\Lambda, 0)\left(Q^T\right)^{-1}M^{\dagger}Q^{-1}\begin{pmatrix} V^T \\ \tilde{V}^T \end{pmatrix},$$

which comes to be the right hand side of (A.7). Now it is easy to verify (A.8). $\square$

**Lemma 5.** *If*

$$K = \begin{pmatrix} P & Q \\ Q^T & R \end{pmatrix} \succeq 0,$$

*then*

$$(u^T, v^T)\begin{pmatrix} P & Q \\ Q^T & R \end{pmatrix}\begin{pmatrix} u \\ v \end{pmatrix} \geq \max\left(u^T\left(P - QR^{\dagger}Q^T\right)u, \ v^T\left(R - Q^T P^{\dagger}Q\right)v\right). \tag{A.9}$$

*Moreover, for* $0 \leq \lambda \leq 1$*, the following two statements are equivalent:*

$$P - QR^{\dagger}Q^T \succeq \lambda P, \tag{A.10}$$

$$R - Q^T P^{\dagger}Q \succeq \lambda R. \tag{A.11}$$

*And if* (A.10) *and* (A.11) *hold, then by* (A.9) *we have*

$$(u^T, v^T)\begin{pmatrix} P & Q \\ Q^T & R \end{pmatrix}\begin{pmatrix} u \\ v \end{pmatrix} \geq \max\left(\lambda u^T Pu, \ \lambda v^T Rv\right)$$

$$\geq \lambda \max\left(\lambda_{\min}(P)\|u\|_2^2, \ \lambda_{\min}(R)\|v\|_2^2\right) \geq \frac{\lambda}{1/\lambda_{\min}(P) + 1/\lambda_{\min}(R)}\left\|\begin{pmatrix} u \\ v \end{pmatrix}\right\|_2^2. \tag{A.12}$$

*Proof.* Theorem 1.19 in [Zha06] tells that $PP^\dagger Q = Q$, so it is easy to verify

$$K = \begin{pmatrix} I & 0 \\ Q^T P^\dagger & I \end{pmatrix} \begin{pmatrix} P & 0 \\ 0 & R - Q^T P^\dagger Q \end{pmatrix} \begin{pmatrix} I & P^\dagger Q \\ 0 & I \end{pmatrix}.$$

Thus

$$\left(u^T, v^T\right) K \begin{pmatrix} u \\ v \end{pmatrix} = \begin{pmatrix} u + P^\dagger R v \\ v \end{pmatrix}^T \begin{pmatrix} P & 0 \\ 0 & R - Q^T P^\dagger Q \end{pmatrix} \begin{pmatrix} u + P^\dagger R v \\ v \end{pmatrix} \geq v^T \left(R - Q^T P^\dagger Q\right) v.$$

Similarly we can obtain another inequality.

If (A.10) holds, then

$$P^{\dagger 1/2} Q R^{\dagger 1/2} \cdot R^{\dagger 1/2} Q^T P^{\dagger 1/2} \preceq (1-\lambda) P^{\dagger 1/2} P P^{\dagger 1/2} \preceq (1-\lambda)I$$
$$\Longrightarrow R^{\dagger 1/2} Q^T P^{\dagger 1/2} \cdot P^{\dagger 1/2} Q R^{\dagger 1/2} \preceq (1-\lambda)I \Longrightarrow R^{1/2} R^{\dagger 1/2} Q^T P^\dagger Q R^{\dagger 1/2} R^{1/2} \preceq (1-\lambda)R.$$

By Theorem 1.19 in [Zha06] we have $QR^{\dagger 1/2}R^{1/2} = Q$, thus $Q^T P^\dagger Q \preceq (1-\lambda)R$, i.e. (A.11) holds. Similarly (A.11) implies (A.10). $\qquad\square$

**Lemma 6.** *Adopt the notation from* (2.9) *to* (2.11). *If* (2.2) *holds, then* (2.6) *holds for* $\lambda_\Sigma = \lambda_H$. *On the other hand, if* (2.6) *holds, then* (2.2) *holds for*

$$\lambda_H = \frac{\lambda_\Sigma \lambda_D^2}{\lambda_D^2 + \left(\nu\Lambda_X^2 + \Lambda_D^2\right)\left(1/\left(\lambda_D^2\right) + 1/\left(\nu\lambda_1^2\right)\right)}.$$

*Proof.* If (2.2) holds, note that

$$H_{(\beta,S),(\beta,S)} = QMQ^T, \text{ where } Q := \begin{pmatrix} I_p & 0 \\ -D_S A^\dagger & I_s \end{pmatrix}, \quad M := \begin{pmatrix} A/\nu & 0 \\ 0 & \Sigma_{S,S} \end{pmatrix}. \tag{A.13}$$

So the left hand side of (2.2) can be written as

$$\begin{pmatrix} \beta - A^\dagger D_S^T \gamma_S \\ \gamma_S \end{pmatrix}^T \begin{pmatrix} A/\nu & 0 \\ 0 & \Sigma_{S,S} \end{pmatrix} \begin{pmatrix} \beta - A^\dagger D_S^T \gamma_S \\ \gamma_S \end{pmatrix}.$$

Taking $\beta = A^\dagger D_S^T \gamma_S \in L$, it becomes $\gamma_S^T \Sigma_{S,S} \gamma_S$, which is not less than $\lambda_H \|\gamma_S\|_2^2$ for any $\gamma_S \in \mathbb{R}^s$. So $\Sigma_{S,S} \succeq \lambda_H I$.

On the other hand, if (2.6) holds, since

$$H_{S,S} - H_{S,\beta} H_{\beta,\beta}^\dagger H_{\beta,S} = \Sigma_{S,S} \succeq \lambda_\Sigma I = \lambda_\Sigma \nu \cdot H_{S,S}, \tag{A.14}$$

by (A.12) we have

$$\left(\beta^T, \gamma_S^T\right) \cdot H_{(\beta,S),(\beta,S)} \cdot \begin{pmatrix} \beta \\ \gamma_S \end{pmatrix} \geq \lambda_\Sigma \nu \cdot \gamma_S^T H_{S,S} \gamma_S = \lambda_\Sigma \|\gamma_S\|_2^2.$$

By Lemma 3, let $\beta = V\delta + \tilde{V}V_1\xi$. By Lemma 5 and (A.14), we know $H_{\beta,\beta} - H_{\beta,S}H_{S,S}^\dagger H_{S,\beta} \succeq \lambda_\Sigma \nu \cdot H_{\beta,\beta}$, and

$$\left(\beta^T, \gamma_S^T\right) \cdot H_{(\beta,S),(\beta,S)} \cdot \begin{pmatrix} \beta \\ \gamma_S \end{pmatrix} \geq \lambda_\Sigma \nu \cdot \beta^T H_{\beta,\beta} \beta$$

$$= \lambda_\Sigma \left(\delta^T, \xi^T\right) \begin{pmatrix} \Lambda^2 + \nu V^T X^* X V & \nu V^T X^* X \tilde{V} V_1 \\ \nu V_1^T \tilde{V}^T X^* X V & \nu V_1^T \tilde{V}^T X^* X \tilde{V} V_1 \end{pmatrix} \begin{pmatrix} \delta \\ \xi \end{pmatrix} := \lambda_\Sigma \left(\delta^T, \xi^T\right) \begin{pmatrix} P & Q \\ Q^T & R \end{pmatrix} \begin{pmatrix} \delta \\ \xi \end{pmatrix}.$$

We have

$$P - QR^\dagger Q^T = \Lambda^2 + \nu V^T X^* (I - U_1 U_1^T) X V \succeq \lambda_D^2 I \succeq \frac{\lambda_D^2}{\nu\Lambda_X^2 + \Lambda_D^2} P.$$

Thus by (A.12),

$$\left(\delta^T, \xi^T\right) \begin{pmatrix} P & Q \\ Q^T & R \end{pmatrix} \begin{pmatrix} \delta \\ \xi \end{pmatrix} \geq \frac{\lambda_D^2}{\nu\Lambda_X^2 + \Lambda_D^2} \cdot \frac{1}{1/\lambda_{\min}(P) + 1/\lambda_{\min}(R)} \left(\|\delta\|_2^2 + \|\xi\|_2^2\right)$$

$$\geq \frac{\lambda_D^2}{\left(\nu\Lambda_X^2 + \Lambda_D^2\right)\left(1/\lambda_D^2 + 1/(\nu\lambda_1^2)\right)} \|\beta\|_2^2.$$

Concluding the results above, we get

$$\left\|\begin{pmatrix}\beta\\\gamma_S\end{pmatrix}\right\|_2^2 = \|\beta\|_2^2 + \|\gamma_S\|_2^2$$

$$\leq \frac{1}{\lambda_\Sigma}\left(1 + \frac{\left(\nu\Lambda_X^2 + \Lambda_D^2\right)\left(1/\lambda_D^2 + 1/\left(\nu\lambda_1^2\right)\right)}{\lambda_D^2}\right)\cdot\left(\beta^T, \gamma_S^T\right)\cdot H_{(\beta,S),(\beta,S)}\cdot\begin{pmatrix}\beta\\\gamma_S\end{pmatrix}.$$

Thus (2.2) holds for

$$\lambda_H = \frac{\lambda_\Sigma\lambda_D^2}{\lambda_D^2 + \left(\nu\Lambda_X^2 + \Lambda_D^2\right)\left(1/\left(\lambda_D^2\right) + 1/\left(\nu\lambda_1^2\right)\right)}.$$

$\square$

**Lemma 7.** *When $\Sigma_{S,S} \succ 0$, we have*

$$H_{S^c,(\beta,S)}H_{(\beta,S),(\beta,S)}^\dagger = \left(-D_{S^c}A^\dagger + \Sigma_{S^c,S}\Sigma_{S,S}^{-1}D_S A^\dagger,\ \Sigma_{S^c,S}\Sigma_{S,S}^{-1}\right). \tag{A.15}$$

*Consequently, for any $\rho \in [-1,1]^s$, we have*

$$\sup_{\rho\in[-1,1]^s}\left\|H_{S^c,(\beta,S)}H_{(\beta,S),(\beta,S)}^\dagger\cdot\begin{pmatrix}0\\\rho\end{pmatrix}\right\|_\infty = \sup_{\rho\in[-1,1]^s}\left\|\Sigma_{S^c,S}\Sigma_{S,S}^{-1}\cdot\rho\right\|_\infty = \left\|\Sigma_{S^c,S}\Sigma_{S,S}^{-1}\right\|_\infty.$$

*Proof.* By (A.13), we know

$$\mathrm{rank}\left(H_{(\beta,S),(\beta,S)}\right) = \mathrm{rank}\left(\begin{pmatrix}A/\nu & 0\\0 & \Sigma_{S,S}\end{pmatrix}\right)$$

$$= \mathrm{rank}(A) + \mathrm{rank}\left(\Sigma_{S,S}\right) = \mathrm{rank}\left(H_{\beta,\beta}\right) + \mathrm{rank}\left(H_{S,S}\right).$$

Then by Theorem 1.21 in [Zha06], we have that

$$H_{(\beta,S),(\beta,S)}^\dagger = \begin{pmatrix}\nu A^\dagger + A^\dagger D_S^T\Sigma_{S,S}^{-1}D_S A^\dagger & A^\dagger D_S^T\Sigma_{S,S}^{-1}\\\Sigma_{S,S}^{-1}D_S A^\dagger & \Sigma_{S,S}^{-1}\end{pmatrix}.$$

By $H_{S^c,(\beta,S)} = (-D_{S^c}/\nu, 0)$ and $-D_{S^c}A^\dagger D_S/\nu = \Sigma_{S^c,S}$, we are done. $\square$

# B    Proof of Theorem 1

*Proof of Theorem 1.* By definition, we have $\mathrm{IC}_0 \geq \|\Omega^S\mathrm{sign}(D_S\beta^\star)\|_\infty \geq \mathrm{IC}_1$. Now we prove $\mathrm{IRR}(0)$ exists and $\mathrm{IRR}(0) = \mathrm{IC}_0$. Let $M := \Lambda^{-1}V^TX^*(I - U_1U_1^T)XV\Lambda^{-1}$. When $\nu$ is small, by (A.8),

$$\nu\Sigma = I - U\Lambda B^{-1}\Lambda U^T = I - U\left(I + \nu M\right)^{-1}U^T$$

$$= I - U\left(I - \nu M + O\left(\nu^2\right)\right)U^T = I - UU^T + \nu UMU^T + O\left(\nu^2\right)$$

$$\implies \nu\Sigma_{S^c,S} = -U_{S^c}U_S^T + \nu U_{S^c}MU_S^T + O\left(\nu^2\right),\ \nu\Sigma_{S,S} = I - U_SU_S^T + \nu U_SMU_S^T + O\left(\nu^2\right).$$

Let $F := I - U_SU_S^T$ and $F = U'\Lambda'U'^T$ be the "compact" eigendecomposition of $F$ ($\Lambda' \succ 0$). Let $G := U_SMU_S^T$. Suppose $(U', \tilde{U}')$ is an orthogonal square matrix, and

$$K = \begin{pmatrix}K_1 & K_2\\K_2^T & K_3\end{pmatrix} := \begin{pmatrix}U'^T\\\tilde{U}'^T\end{pmatrix}G\left(U', \tilde{U}'\right).$$

By $F + \nu G \succ 0$, we have $K_3 \succ 0$. Now

$$F + \nu G = \left(U', \tilde{U}'\right)\begin{pmatrix}\Lambda' + \nu K_1 & \nu K_2\\\nu K_2^T & \nu K_3\end{pmatrix}\begin{pmatrix}U'^T\\\tilde{U}'^T\end{pmatrix}.$$

Define $Q_\nu = K_3 - \nu K_2^T(\Lambda' + \nu K_1)^{-1}K_2$, $R_\nu = K_2^T(\Lambda' + \nu K_1)^{-1}$, and we can calculate

$$(F + \nu G)^{-1} = \left(U', \tilde{U}'\right)\begin{pmatrix}(\Lambda' + \nu K_1)^{-1} + \nu R_\nu^TQ_\nu^{-1}R_\nu & -R_\nu^TQ_\nu^{-1}\\-Q_\nu^{-1}R_\nu & Q_\nu/\nu\end{pmatrix}\begin{pmatrix}U'^T\\\tilde{U}'^T\end{pmatrix}.$$

Note that $Q_\nu \to K_3$, $R_\nu \to K_2^T \Lambda'^{-1}$, and note that

$$U_{S^c}^T U_{S^c} U_S^T \tilde{U}' = \left(I - U_S^T U_S\right) U_S^T \tilde{U}' = U_S^T \left(I - U_S U_S^T\right) \tilde{U}' = U_S^T U' \Lambda' \cdot U'^T \tilde{U}' = 0$$

$$\implies \left(U_{S^c} U_S^T \tilde{U}'\right)^T U_{S^c} U_S^T \tilde{U}' = 0 \implies U_{S^c} U_S^T \tilde{U}' = 0. \quad \text{(B.1)}$$

Combining it with the representation of $(F + \nu G)^{-1}$,

$$-U_{S^c} U_S^T \Sigma_{S,S}^{-1} \doteq U_{S^c} U_S^T (F + \nu G)^{-1}$$

$$= -\left(U_{S^c} U_S^T U', 0\right) \begin{pmatrix} (\Lambda' + \nu K_1)^{-1} + \nu R_\nu^T Q_\nu^{-1} R_\nu & -R_\nu^T Q_\nu^{-1} \\ -Q_\nu^{-1} R_\nu & \star \end{pmatrix} \begin{pmatrix} U'^T \\ \tilde{U}'^T \end{pmatrix}$$

$$\to \left(-U_{S^c} U_S^T U' \Lambda'^{-1}, U_{S^c} U_S^T U' \Lambda'^{-1} K_2 K_3^{-1}\right) \begin{pmatrix} U'^T \\ \tilde{U}' \end{pmatrix} = -U_{S^c} U_S^T U' \Lambda'^{-1} \left(U'^T - K_2 K_3^{-1} \tilde{U}'^T\right).$$

Besides,

$$\nu U_{S^c} M U_S^T \Sigma_{S,S}^{-1} \doteq U_{S^c} M U_S^T \cdot \nu \left(F + \nu G\right)^{-1} \to U_{S^c} M U_S^T \tilde{U}' K_3^{-1} \tilde{U}'^T.$$

So when $\nu \to 0$,

$$\Sigma_{S^c,S} \Sigma_{S,S}^{-1} \to -U_{S^c} U_S^T U' \Lambda'^{-1} \left(U'^T - K_2 K_3^{-1} \tilde{U}'^T\right) + U_{S^c} M U_S^T \tilde{U}' K_3^{-1} \tilde{U}'^T$$

$$= -U_{S^c} U_S^T U' \Lambda'^{-1} U'^T + U_{S^c} \left(U_S^T U' \Lambda'^{-1} U'^T U_S + I\right) M U_S^T \tilde{U}' K_3^{-1} \tilde{U}'^T$$

$$= -D_{S^c} V \Lambda^{-1} U_S^T U' \Lambda'^{-1} U'^T + D_{S^c} V \Lambda^{-1} \left(I + U_S^T U' \Lambda'^{-1} U'^T U_S\right) M U_S^T \tilde{U}' K_3^{-1} \tilde{U}'^T.$$

The infinity norm of the right hand side is $\text{IRR}(0)$. On the other hand,

$$\text{IC}_0 = \left\| D_{S^c} \left(D_{S^c}^T D_{S^c}\right)^\dagger \left(X^* X W \left(W^T X^* X W\right)^\dagger W^T - I\right) D_S^T \right\|_\infty.$$

In order to prove $\text{IRR}(0) = \text{IC}_0$, it suffices to show

$$\left(X^* X W \left(W^T X^* X W\right)^\dagger W^T - I\right) D_S^T = -D_{S^c}^T D_{S^c} V \Lambda^{-1} U_S^T U' \Lambda'^{-1} U'^T$$

$$+ D_{S^c}^T D_{S^c} V \Lambda^{-1} \left(I + U_S^T U' \Lambda'^{-1} U'^T U_S\right) M U_S^T \tilde{U}' K_3^{-1} \tilde{U}'^T.$$

The first term of the right hand side is

$$-V \Lambda U_{S^c}^T U_{S^c} U_S^T U' \Lambda'^{-1} U'^T = -V \Lambda \left(I - U_S^T U_S\right) U_S^T U' \Lambda'^{-1} U'^T$$

$$= -V \Lambda U_S^T \left(I - U_S U_S^T\right) U' \Lambda'^{-1} U'^T = -V \Lambda U_S^T U' \Lambda' U'^T U' \Lambda'^{-1} U'^T = -D_S^T U' U'^T,$$

while by the fact that

$$\left(I - U_S^T U_S\right) \left(I + U_S^T U' \Lambda'^{-1} U'^T U_S\right)$$

$$= I - U_S^T U_S + U_S^T U' \Lambda'^{-1} U'^T U_S - U_S^T U_S U_S^T U' \Lambda'^{-1} U'^T U_S$$

$$= I - U_S^T U_S + U_S^T \left(I - U_S U_S^T\right) U' \Lambda'^{-1} U'^T U_S$$

$$= I - U_S^T U_S + U_S^T U' U'^T U_S = I - U_S^T \tilde{U}' \tilde{U}'^T U_S,$$

the second term becomes

$$V \Lambda U_{S^c}^T U_{S^c} \left(I + U_S^T U' \Lambda'^{-1} U'^T U_S\right) M U_S^T \tilde{U}' K_3^{-1} \tilde{U}'^T$$

$$= V \Lambda \left(I - U_S^T U_S\right) \left(I + U_S^T U' \Lambda'^{-1} U'^T U_S\right) M U_S^T \tilde{U}' K_3^{-1} \tilde{U}'^T$$

$$= V \Lambda \left(I - U_S^T \tilde{U}' \tilde{U}'^T U_S\right) M U_S^T \tilde{U}' K_3^{-1} \tilde{U}'^T$$

$$= V \Lambda M U_S^T \tilde{U}' K_3^{-1} \tilde{U}'^T - V \Lambda U_S^T \tilde{U}' \cdot \tilde{U}'^T U_S M U_S^T \tilde{U}' \cdot K_3^{-1} \tilde{U}'^T$$

$$= X^* \left(I - U_1 U_1^T\right) X V \Lambda^{-1} U_S^T \tilde{U}' K_3^{-1} \tilde{U}' - D_S^T \tilde{U}' \cdot K_3 \cdot K_3^{-1} \tilde{U}'^T$$

$$= X^* \left(I - U_1 U_1^T\right) X V \Lambda^{-1} U_S^T \tilde{U}' K_3^{-1} \tilde{U}'^T - D_S^T \tilde{U}' \tilde{U}'^T.$$

So it suffices to show

$$X^* X W \left(W^T X^* X W\right)^\dagger W^T D_S^T = X^* \left(I - U_1 U_1^T\right) X V \Lambda^{-1} U_S^T \tilde{U}' K_3^{-1} \tilde{U}'^T,$$

which is equivalent to

$$X^* \left(X W W^T X^*\right)^\dagger X W W^T D_S^T = X^* \left(I - U_1 U_1^T\right) X V \Lambda^{-1} U_S^T \tilde{U}' K_3^{-1} \tilde{U}'^T. \tag{B.2}$$

First we prove

$$\ker\left(U_{S^c}\right) = \mathrm{Im}\left(U_S^T \tilde{U}'\right). \tag{B.3}$$

In fact, by (B.1) we have $\mathrm{Im}(U_S^T \tilde{U}') \subseteq \ker(U_{S^c})$. For any $\zeta \in \ker(U_{S^c})$, we have $(I - U_S^T U_S)\zeta = U_{S^c}^T U_{S^c}\zeta = 0$. Let

$$\zeta = U_S^T \zeta_1 + \zeta_2, \ \zeta_2 \in \ker(U_S),$$

then

$$0 = (I - U_S^T U_S)(U_S^T \zeta_1 + \zeta_2) = \zeta_2 + (I - U_S^T U_S) U_S^T \zeta_1 = \zeta_2 + U_S^T (I - U_S U_S^T)\zeta_1,$$

which implies $\zeta_2 \in \mathrm{Im}(U_S^T)$. But $\zeta_2 \in \ker(U_S)$, then $\zeta_2 = 0$, and $0 = (I - U_S^T U_S) U_S^T \zeta_1 = U_S^T (I - U_S U_S^T)\zeta_1 = U_S^T U' \Lambda' U'^T \zeta_1$. Assume that $\zeta_1 = U' \zeta_3 + \tilde{U}' \tilde{\zeta}_3$, then $U_S^T U' \Lambda' \zeta_3 = 0$. Thus

$$0 = U_S U_S^T U' \Lambda' \zeta_3 = \left(I - U' \Lambda' U'^T\right) U' \Lambda' \zeta_3 = U' \Lambda' \left(I - \Lambda'\right) \zeta_3 \implies \left(I - \Lambda'\right) \zeta_3 = 0$$

$$\implies U_S U_S^T U' \zeta_3 = U' \left(I - \Lambda'\right) \zeta_3 = 0 \implies \left(U_S^T U' \zeta_3\right)^T U_S^T U' \zeta_3 = 0 \implies U_S^T U' \zeta_3 = 0$$

$$\implies \beta = U_S^T \zeta_1 = U_S^T U' \zeta_3 + U_S^T \tilde{U}' \tilde{\zeta}_3 = U_S^T \tilde{U}' \tilde{\zeta}_3 \in \mathrm{Im}\left(U_S^T \tilde{U}'\right).$$

So (B.3) holds. Now for any $\beta \in \mathbb{R}^p$, let $\beta = V\delta + \tilde{V}\tilde{\delta}$, then $\beta \in \ker(D_{S^c})$ if and only if $U_{S^c} \Lambda \delta = 0$, which means $\delta \in \Lambda^{-1} \ker(U_{S^c}) = \mathrm{Im}(\Lambda^{-1} U_S^T \tilde{U}')$. So

$$\ker\left(D_{S^c}\right) = \mathrm{Im}\left(J\right) + \mathrm{Im}\left(\tilde{V}\right), \ \text{where } J := V \Lambda^{-1} U_S^T \tilde{U}'.$$

Since $\tilde{V}^T V = 0$, the linear subspaces spanned by $J$ and $\tilde{V}$ are orthogonal, and we have

$$W W^T = J \left(J^T J\right)^\dagger J^T + \tilde{V} \tilde{V}^T.$$

Noting $\tilde{V}^T V = 0$, $\tilde{V}^T X^*(I - U_1 U_1^T) = 0$, we have

$$X^* \left(X W W^T X^*\right)^\dagger X W W^T D_S^T \tilde{U}' \tilde{U}'^T K_3$$

$$= X^* \left(X W W^T X^*\right)^\dagger X J \left(J^T J\right)^\dagger J^T V \Lambda U_S^T \tilde{U}' \tilde{U}'^T K_3$$

$$= X^* \left(X W W^T X^*\right)^\dagger X J \left(J^T J\right)^\dagger \cdot \tilde{U}'^T U_S U_S^T \tilde{U}' \cdot \tilde{U}'^T U_S M U_S^T \tilde{U}'$$

$$= X^* \left(X W W^T X^*\right)^\dagger X J \left(J^T J\right)^\dagger \cdot \tilde{U}'^T U_S M U_S^T \tilde{U}'$$

$$= X^* \left(X W W^T X^*\right)^\dagger X J \left(J^T J\right)^\dagger J^T X^* \left(I - U_1 U_1^T\right) X V \Lambda^{-1} U_S^T \tilde{U}'$$

$$= X^* \left(X W W^T X^*\right)^\dagger \left(X W W^T X^*\right) \left(I - U_1 U_1^T\right) X J.$$

Since $(X W W^T X^*)^\dagger (X W W^T X^*)$ is the projection matrix onto the linear subspace $\mathrm{Im}(XW) = \mathrm{Im}(X\tilde{V}) + \mathrm{Im}(XJ) = \mathrm{Im}(U_1) + \mathrm{Im}(XJ)$, and $(I - U_1 U_1^T)XJ = XJ - U_1 \cdot U_1^T X J$ lies in this subspace, the last term above becomes $X^* \left(I - U_1 U_1^T\right) X J$. Therefore, we get

$$X^* \left(X W W^T X^*\right)^\dagger X W W^T D_S^T \tilde{U}' K_3 = X^* \left(I - U_1 U_1^T\right) X J$$

$$\iff X^* \left(X W W^T X^*\right)^\dagger X W W^T D_S^T \tilde{U}' = X^* \left(I - U_1 U_1^T\right) X V \Lambda^{-1} U_S^T \tilde{U}' K_3^{-1}.$$

Now to prove (B.2), it suffices to show

$$X^* \left(X W W^T X^*\right)^\dagger X W W^T D_S^T \left(I - \tilde{U}' \tilde{U}'^T\right) = 0 \impliedby W W^T D_S^T U' U'^T = 0$$

$$\impliedby J \left(J^T J\right)^\dagger J^T D_S^T U' = 0 \impliedby J^T D_S^T U' = 0 \impliedby \tilde{U}'^T U_S \Lambda^{-1} V^T \cdot V \Lambda U_S^T U' = 0$$

$$\impliedby \tilde{U}'^T U_S U_S^T U' = 0 \impliedby \tilde{U}'^T \left(I - U' \Lambda' U'^T\right) U' = 0,$$

which is surely true since $\tilde{U}'^T U' = 0$. Then $\mathrm{IRR}(0) = \mathrm{IC}_0$ is proved.

Now we turn to $\mathrm{IRR}(\infty)$. Let $M = U'' \Lambda'' U''^T$ be the compact eigendecomposition of $M$, and $(U'', \tilde{U}'')$ is an orthogonal square matrix. Then

$$
\nu \Sigma = I - U \left( I + \nu M \right)^{-1} U^T
$$

$$
= I - U \left( U'', \tilde{U}'' \right) \left( \begin{pmatrix} U''^T \\ \tilde{U}''^T \end{pmatrix} (I + \nu M) \left( U'', \tilde{U}'' \right) \right)^{-1} \begin{pmatrix} U''^T \\ \tilde{U}''^T \end{pmatrix} U^T
$$

$$
= I - U \left( U'', \tilde{U}'' \right) \begin{pmatrix} I + \nu \Lambda'' & 0 \\ 0 & I \end{pmatrix}^{-1} \begin{pmatrix} U''^T \\ \tilde{U}''^T \end{pmatrix} U^T
$$

$$
= I - U U'' \left( I + \nu \Lambda'' \right)^{-1} U''^T U^T - U \tilde{U}'' \tilde{U}''^T U^T \to I - U \tilde{U}'' \tilde{U}''^T U^T
$$

when $\nu \to +\infty$. Besides, $\nu \Sigma_{S,S} \to I - U_S \tilde{U}'' \tilde{U}''^T U_S^T$, and this limit $\succeq \nu \Sigma_{S,S} \succ 0$ for any $\nu > 0$. Thus $\Sigma_{S^c,S} \Sigma_{S,S}^{-1}$ has limit when $\nu \to +\infty$.

Now we study when $\mathrm{IRR}(\infty) = 0$. The underlying existence of $\Sigma_{S,S}^{-1}$ requires $\Sigma_{S,S} \succ 0$, which is equivalent to $\ker(D_{S^c}) \cap \ker(X) \subseteq \ker(D_S)$ by Lemma 2. Let

$$
D_S^T = X^T C_1 + D_{S^c}^T C_2, \text{ which implies } U_S^T = \Lambda^{-1} V^T X^T C_1 + U_{S^c}^T C_2.
$$

Then $0 = \tilde{V}^T D_S^T = \tilde{V}^T X^T C_1 + 0 = \sqrt{n} V_1 \Lambda_1 U_1^T C_1$, which implies $U_1^T C_1 = 0$. So for $N = \Lambda^{-1} V^T X^T (I - U_1 U_1^T)/\sqrt{n}$, we have

$$
N C_1 = \Lambda^{-1} V^T X^T C_1 / \sqrt{n}.
$$

Then $\mathrm{IRR}(\infty) = 0 \iff -U_{S^c} \tilde{U}'' \tilde{U}''^T U_S^T = 0 \iff -U_{S^c}(I - M M^\dagger) U_S^T = 0$. By $M = N N^T$, the equation is further equivalent to

$$
-U_{S^c} \left( I - N N^\dagger \right) U_S^T = 0 \iff -U_{S^c} \left( I - N N^\dagger \right) \left( \Lambda^{-1} V^T X C_1 + U_{S^c}^T C_2 \right) = 0
$$

$$
\iff -U_{S^c} \left( I - N N^\dagger \right) \left( \sqrt{n} N C_1 + U_{S^c}^T C_2 \right) = 0
$$

$$
\iff -U_{S^c} \left( I - N N^\dagger \right) U_{S^c}^T C_2 = 0 \iff C_2^T U_{S^c} \left( I - N N^\dagger \right) \cdot \left( I - N N^\dagger \right) U_{S^c}^T C_2 = 0
$$

$$
\iff \left( I - N N^\dagger \right) U_{S^c}^T C_2 = 0 \iff \mathrm{Im}(U_{S^c}^T C_2) \subseteq \mathrm{Im}(N).
$$

It suffices to show that the last property holds if and only if $\ker(X) \subseteq \ker(D_S)$ or, equivalently, $\mathrm{Im}(D_S^T) \subseteq \mathrm{Im}(X^T)$. In fact, if $\mathrm{Im}(D_S^T) \subseteq \mathrm{Im}(X^T)$, then $C_2$ can be set $0$ in the beginning, and $\mathrm{Im}(U_{S^c}^T C_2) = \mathrm{Im}(0) \subseteq \mathrm{Im}(N)$. If $\mathrm{Im}(U_{S^c}^T C_2) \subseteq \mathrm{Im}(N)$, let $U_{S^c}^T C_2 = N C_3$, then

$$
D_{S^c}^T C_2 = V \Lambda U_{S^c}^T C_2 = V V^T X^T \left( I - U_1 U_1^T \right) C_3 / \sqrt{n}
$$

$$
= \left( V V^T + \tilde{V} \tilde{V}^T \right) X^T \left( I - U_1 U_1^T \right) C_3 / \sqrt{n} = X^T \left( I - U_1 U_1^T \right) C_3 / \sqrt{n},
$$

and hence $D_S^T = X^T C_1 + D_{S^c}^T C_2 = X^T (C_1 + (I - U_1 U_1^T) C_3 / \sqrt{n})$, which implies $\mathrm{Im}(D_S^T) \subseteq \mathrm{Im}(X^T)$. We have finished the proof of that $\mathrm{IRR}(\infty) = 0$ if and only if $\ker(X) \subseteq \ker(D_S)$. $\qquad\square$

## C   Split Linearized Inverse Scale Space (Split LBISS) as the Limit Dynamics of Split LBI

Now we focus on a differential inclusion called *Split Linearized Bregman Inverse Scale Space (Split LBISS)*, the limit dynamics of Split LBI when the step size $\alpha \to 0$. This dynamics helps us understand the behavior of Split LBI, and the proof on sign consistency as well as $\ell_2$ consistency of Split LBISS can be rewritten into a discrete version then applied to Split LBI with slight modifications.

First, noting that

$$
\rho \in \partial \left\| \gamma \right\|_1, \; z = \rho + \gamma/\kappa \qquad \Longleftrightarrow \qquad \gamma = \kappa \mathcal{S}(z, 1), \; \rho = z - \mathcal{S}(z, 1), \tag{C.1}
$$

we have an equivalent form of Split LBI as follows.

$$\beta_{k+1}/\kappa = \beta_k/\kappa - \alpha\nabla_\beta\ell\left(\beta_k,\gamma_k\right),\tag{C.2a}$$

$$\rho_{k+1} + \gamma_{k+1}/\kappa = \rho_k + \gamma_k/\kappa - \alpha\nabla_\gamma\ell\left(\beta_k,\gamma_k\right),\tag{C.2b}$$

$$\rho_k \in \partial\left\|\gamma_k\right\|_1,\tag{C.2c}$$

where $\rho_0 = \gamma_0 = 0 \in \mathbb{R}^m$, $\beta_0 = 0 \in \mathbb{R}^p$. By letting $\rho(k\alpha) = \rho_k$, $\gamma(k\alpha) = \gamma_k$, $\beta(k\alpha) = \beta_k$ and $\alpha \to 0$, the iteration above can be viewed as a forward Euler discretization to the following inclusion called *Split Linearized Bregman Inverse Scale Space (Split LBISS)*.

$$\dot{\beta}(t)/\kappa = -\nabla_\beta\ell\left(\beta(t),\gamma(t)\right) = -X^*\left(X\beta(t)-y\right) - D^T\left(D\beta(t)-\gamma(t)\right)/\nu,\tag{C.3a}$$

$$\dot{\rho}(t) + \dot{\gamma}(t)/\kappa = -\nabla_\gamma\ell\left(\beta(t),\gamma(t)\right) = -\left(\gamma(t)-D\beta(t)\right)/\nu,\tag{C.3b}$$

$$\rho(t) \in \partial\left\|\gamma(t)\right\|_1,\tag{C.3c}$$

where $\rho(t),\beta(t),\gamma(t)$ are right continuously differentiable, with $\dot{\rho}(t),\dot{\beta}(t),\dot{\gamma}(t)$ denoting the right derivatives in $t$ of $\rho(t),\beta(t),\gamma(t)$ respectively, and $\rho(0) = \gamma(0) = 0 \in \mathbb{R}^m$, $\beta(0) = 0 \in \mathbb{R}^p$.

The following inclusion called *Split Inverse Scale Space (Split ISS)* can be viewed as the limit of Split LBISS when $\kappa \to +\infty$.

$$0 = -\nabla_\beta\ell\left(\beta(t),\gamma(t)\right) = -X^*\left(X\beta(t)-y\right) - D^T\left(D\beta(t)-\gamma(t)\right)/\nu,\tag{C.4a}$$

$$\dot{\rho}(t) = -\nabla_\gamma\ell\left(\beta(t),\gamma(t)\right) = -\left(\gamma(t)-D\beta(t)\right)/\nu,\tag{C.4b}$$

$$\rho(t) \in \partial\left\|\gamma(t)\right\|_1,\tag{C.4c}$$

where $\rho(t)$ is right continuously differentiable, $\beta(t),\gamma(t)$ are right continuous, and $\rho(0) = \gamma(0) = 0 \in \mathbb{R}^m$, $\beta(0) = 0 \in \mathbb{R}^p$. Besides, we require "$\beta(t) \in L$", since replacing $\beta(t)$ with "the projection of $\beta(t)$ onto $L$" does not disturb (C.4a) to (C.4c). (C.4) coincides with the differential inclusion proposed in Chapter 8 of [Moe12], there the authors introduced it from another aspect.

The following propositions establish the solution existence and uniqueness of Split (LB)ISS, in almost the same way as [Osh+16].

**Proposition 1** (Solution existence and uniqueness for Split (LB)ISS).

1. *As for Split ISS (C.4), assume that $\rho(t)$ is right continuously differentiable and $\beta(t),\gamma(t)$ is right continuous. Then a solution exists for $t \geq 0$, with piecewise linear $\rho(t)$ and piecewise constant $\beta(t),\gamma(t)$. Besides, $\rho(t)$ is unique. If additionally $\Sigma_{S(t),S(t)} \succ 0$ for $0 \leq t \leq \tau$, where $\Sigma$ is defined in (2.5) and $S(t) := \mathrm{supp}(\gamma(t))$, then $\beta(t),\gamma(t)$ are unique for $0 \leq t \leq \tau$.*

2. *As for Split LBISS (C.3), assume that $\rho(t),\beta(t)$ are right continuously differentiable. Then a solution exists for $t \geq 0$.*

*Proof of proposition 1.* For Split ISS, by (C.4a) and the fact that $\beta(t) \in L = \mathrm{Im}(X^T) + \mathrm{Im}(D^T) = \mathrm{Im}(A) = \mathrm{Im}(A^\dagger)$, we can solve $\beta(t) = A^\dagger(\nu X^*y + D^T\gamma(t))$ which is determined by $\gamma(t)$. Plugging it into (C.4b) we have

$$\dot{\rho}(t) + \dot{\gamma}(t)/\kappa = -\Sigma\gamma(t) + DA^\dagger X^*y.$$

Taking $M = I_{p+m} - (\sqrt{\nu/n}X^T, D^T)^\dagger(\sqrt{\nu/n}X^T, D^T)$ in Theorem 1.19 in [Zha06] leads to

$$DA^\dagger X^* = \Sigma\Sigma^\dagger\left(DA^\dagger X^*\right) = \Sigma^{1/2}\Sigma^{\dagger 1/2}\left(DAX^*\right).$$

The inclusion becomes

$$\dot{\rho}(t) + \dot{\gamma}(t)/\kappa = -\Sigma^{1/2}\left(\Sigma^{1/2}\gamma(t) - \Sigma^{\dagger 1/2}DA^\dagger X^*y\right),$$

which is a standard ISS (on $\gamma(t)$) and has been sufficiently discussed in [Osh+16] (let $X,y$ in that paper take $\sqrt{n}\Sigma^{1/2}$ and $\sqrt{n}\Sigma^{\dagger 1/2}DA^\dagger X^*y$ in this paper). Specifially, there exists a solution with piecewise linear $\rho(t)$ and piecewise constant $\beta(t),\gamma(t)$. Besides, $\rho(t)$ is unique. If additionally, when $\Sigma_{S(t),S(t)} \succ 0$, we have that $\Sigma_{\cdot,S(t)}$ has full column rank, and $\gamma(t)$ (hence $\beta(t)$) is unique.

For Split LBISS, letting $z(t) = \rho(t) + \gamma(t)/\kappa$ and noting (C.1), the Split LBISS (C.3) is equivalent to

$$\begin{pmatrix}\dot{\beta}(t)\\\dot{z}(t)\end{pmatrix} = -\begin{pmatrix}-\kappa X^*\left(X\beta(t)-y\right) - \kappa D^T\left(D\beta(t) - \kappa\mathcal{S}(z(t),1)\right)/\nu\\-\left(\kappa\mathcal{S}(z(t),1) - D\beta(t)\right)/\nu\end{pmatrix}.$$

The Picard-Lindelöf Theorem implies that this ODE has a unique solution $(\beta(t),z(t))$, so there exists a unique solution to the Split LBISS (C.3). $\qquad\square$

Besides, for the loss function defined in (1.3), we have the following property.

**Proposition 2** (Non-increasing $\ell$ along the solutions of Split (LB)ISS and LBI)**.**

1. *For a solution $(\rho(t), \beta(t), \gamma(t))$ of Split ISS (C.4), $\ell(\beta(t), \gamma(t))$ is non-increasing in $t$.*

2. *For a solution $(\rho(t), \beta(t), \gamma(t))$ of Split LBISS (C.3), $\ell(\beta(t), \gamma(t))$ is non-increasing in $t$.*

3. *For a solution $(\rho_k, \beta_k, \gamma_k)$ of Split LBI (C.2), $\ell(\beta_k, \gamma_k)$ is non-increasing in $k$, if*

$$\kappa\alpha\|H\|_2 \leq 2. \tag{C.5}$$

*Moreover, one can prove $\|H\|_2 \leq 2\left(1 + \nu\Lambda_X^2 + \Lambda_D^2\right)/\nu$, so (C.5) holds if*

$$\kappa\alpha \leq \nu/(1 + \nu\Lambda_X^2 + \Lambda_D^2). \tag{C.6}$$

*Proof of proposition 2.* For Split ISS, one can easily imitates the technique in the proof of Theorem 2.1 in [Osh+16] to show that $(\beta(t), \gamma(t))$ is the solution of the following optimization problem.

$$\min_{\beta,\gamma} \quad \ell(\beta(t), \gamma(t))$$

$$\text{subject to} \quad \begin{cases} \gamma_j \geq 0, & \text{if } \rho_j(t) = 1, \\ \gamma_j \leq 0, & \text{if } \rho_j(t) = -1, \\ \gamma_j = 0, & \text{if } \rho_j(t) \in (-1, 1). \end{cases} \tag{C.7}$$

for any $t > 0$, due to the continuity of $\rho(\cdot)$, there is a small neighborhood of $t$, on which every $\tau$ satisfies

$$\begin{cases} \rho_j(\tau) > -1 \text{ hence } \gamma_j(\tau) \geq 0, & \text{if } \rho_j(t) = 1, \\ \rho_j(\tau) < 1 \text{ hence } \gamma_j(\tau) \geq 0, & \text{if } \rho_j(t) = -1, \\ \rho_j(\tau) \in (-1, 1) \text{ hence } \gamma_j(\tau) = 0, & \text{if } \rho_j(t) \in (-1, 1). \end{cases}$$

That is to say, $(\beta(\tau), \gamma(\tau))$ satisfies the constraints in (C.7), so the value of $\ell(\beta(\tau), \gamma(\tau))$ is not less than $\ell(\beta(t), \gamma(t))$, namely the minimum of (C.7). This implies that any $t \geq 0$ is a local minimal point of a right continuous function $\ell(\beta(\cdot), \gamma(\cdot))$. Then by standard techniques in mathematical analysis, we have that $\ell(\beta(t), \gamma(t))$ is non-increasing.

For Split LBISS, by (C.3c), we have $\dot{\gamma}_j(t) \cdot \dot{\rho}_j(t) \equiv 0$ for each $j$, so $\ell$ is non-increasing since

$$\frac{\mathrm{d}}{\mathrm{d}t}\ell(\beta(t), \gamma(t)) = \left\langle \begin{pmatrix} \dot{\beta}(t) \\ \dot{\gamma}(t) \end{pmatrix}, \begin{pmatrix} \nabla_\beta \ell(\beta(t), \gamma(t)) \\ \nabla_\gamma \ell(\beta(t), \gamma(t)) \end{pmatrix} \right\rangle$$

$$= \left\langle \begin{pmatrix} \dot{\beta}(t) \\ \dot{\gamma}(t) \end{pmatrix}, \begin{pmatrix} -\dot{\beta}(t)/\kappa \\ -\dot{\rho}(t) - \dot{\gamma}(t)/\kappa \end{pmatrix} \right\rangle = \frac{1}{\kappa}\left\| \begin{pmatrix} \dot{\beta}(t) \\ \dot{\gamma}(t) \end{pmatrix} \right\|_2^2 \leq 0.$$

For Split LBI, noting $(\rho_{k+1} - \rho_k)(\gamma_{k+1} - \gamma_k) = \|\rho_{k+1}\|_1 - \langle \rho_{k+1}, \gamma_k \rangle + \|\gamma_{k+1}\|_1 - \langle \rho_k, \gamma_{k+1} \rangle \geq 0$, we have

$$-\alpha\nabla\ell(\beta_k, \gamma_k)^T \begin{pmatrix} \beta_{k+1} - \beta_k \\ \gamma_{k+1} - \gamma_k \end{pmatrix}$$

$$= \left(\begin{pmatrix} 0 \\ \rho_{k+1} - \rho_k \end{pmatrix} + \frac{1}{\kappa}\begin{pmatrix} \beta_{k+1} - \beta_k \\ \gamma_{k+1} - \gamma_k \end{pmatrix}\right)\begin{pmatrix} \beta_{k+1} - \beta_k \\ \gamma_{k+1} - \gamma_k \end{pmatrix} \geq \frac{1}{\kappa}\left\|\begin{pmatrix} \beta_{k+1} - \beta_k \\ \gamma_{k+1} - \gamma_k \end{pmatrix}\right\|_2^2.$$

By $\kappa\alpha\|H\|_2 < 2$, we have

$$\ell(\beta_{k+1}, \gamma_{k+1}) - \ell(\beta_k, \gamma_k)$$

$$= \nabla\ell(\beta_k, \gamma_k)^T \begin{pmatrix} \beta_{k+1} - \beta_k \\ \gamma_{k+1} - \gamma_k \end{pmatrix} + \frac{1}{2}\left(\beta_{k+1}^T - \beta_k^T, \gamma_{k+1}^T - \gamma_k^T\right)H\begin{pmatrix} \beta_{k+1} - \beta_k \\ \gamma_{k+1} - \gamma_k \end{pmatrix}$$

$$\leq -\frac{1}{\kappa\alpha}\left\|\begin{pmatrix} \beta_{k+1} - \beta_k \\ \gamma_{k+1} - \gamma_k \end{pmatrix}\right\|_2^2 + \frac{\|H\|_2}{2}\cdot\left\|\begin{pmatrix} \beta_{k+1} - \beta_k \\ \gamma_{k+1} - \gamma_k \end{pmatrix}\right\|_2^2 \leq 0.$$

Moreover, it is easy to verify that

$$
\left(\beta^T, \gamma^T\right) H \begin{pmatrix} \beta \\ \gamma \end{pmatrix} = \frac{1}{n} \|X\beta\|_2^2 + \frac{1}{\nu} \|D\beta - \gamma\|_2^2 \le \frac{2}{n} \|X\beta\|_2^2 + \frac{2}{\nu} \|D\beta\|_2^2 + 2\|\gamma\|_2^2
$$

$$
\le \frac{2\left(1 + \nu\Lambda_X^2 + \Lambda_D^2\right)}{\nu} \left\| \begin{pmatrix} \beta \\ \gamma \end{pmatrix} \right\|_2^2 \quad \left( \begin{pmatrix} \beta \\ \gamma \end{pmatrix} \in \mathbb{R}^{m+p} \right),
$$

$$
\Longrightarrow \|H\|_2 \le \frac{2\left(1 + \nu\Lambda_X^2 + \Lambda_D^2\right)}{\nu}. \quad \text{(C.8)}
$$

$\square$

# D  Oracle Properties: the Key to Prove Consistency

The key to our analysis for Split LBISS and LBI is to deal with the *Oracle* properties, i.e. properties assuming $S$ is known. First, let $(\beta^o, \gamma^o)$ form an *Oracle solution* of minimizing $\ell$, namely

$$
(\beta^o, \gamma^o) \in \arg \min_{\substack{\beta, \gamma \\ \gamma_{S^c} = 0}} \ell(\beta, \gamma). \quad \text{(D.1)}
$$

which implies

$$
\begin{aligned}
\nabla_\beta \ell(\beta^o, \gamma^o) &= X^*(X\beta^o - y) + D^T(D\beta^o - \gamma^o)/\nu = 0, \\
\nabla_{\gamma_S} \ell(\beta^o, \gamma^o) &= (\gamma_S^o - D_S\beta^o)/\nu = 0.
\end{aligned} \quad \text{(D.2)}
$$

Obviously $\ell(P_L\beta^o, \gamma^o) = \ell(\beta^o, \gamma^o)$, thus we can assume that $\beta^o \in L$, and actually

$$
(\beta^o, \gamma^o) \in \arg \min_{\substack{\beta, \gamma \\ \beta \in L, \, \gamma_{S^c} = 0}} \ell(\beta, \gamma). \quad \text{(D.3)}
$$

## D.1  Oracle Dynamics of Split LBISS

Define the *Oracle Dynamics* of Split LBISS (C.3) as

$$
\rho'_{S^c}(t) = \gamma'_{S^c}(t) \equiv 0, \quad \text{(D.4a)}
$$

$$
\dot{\beta}'(t)/\kappa = -X^*(X\beta'(t) - y) - D^T(D\beta'(t) - \gamma'(t))/\nu, \quad \text{(D.4b)}
$$

$$
\dot{\rho}'_S(t) + \dot{\gamma}'_S(t)/\kappa = -(\gamma'_S(t) - D_S\beta'(t))/\nu, \quad \text{(D.4c)}
$$

$$
\rho'_S(t) \in \partial \|\gamma'_S(t)\|_1, \quad \text{(D.4d)}
$$

where $\rho'_S(0) = \gamma'_S(0) = 0 \in \mathbb{R}^s$, $\beta'(0) = 0 \in \mathbb{R}^p$. Besides, we require "$\beta'(t) \in L$". This dynamics can be viewed as an *Oracle* version of Split LBISS (C.3), with $S$ known and $\rho_{S^c}(t)$, $\gamma_{S^c}(t)$ set to be 0. We first expect and prove $(\beta'(t), \gamma'(t))$ converges to $(\beta^o, \gamma^o)$ as $t$ evolves. Let

$$
d_\beta(t) := \beta'(t) - \beta^o, \ d_\gamma(t) := \gamma'(t) - \gamma^o, \ d(t) = \sqrt{\|d_{\gamma,S}(t)\|_2^2 + \|d_\beta(t)\|_2^2}. \quad \text{(D.5)}
$$

Adding (D.2) to (D.4b) and (D.4c), the Oracle Dynamics can be reformulated as

$$
\rho'_{S^c}(t) = \gamma'_{S^c}(t) \equiv 0, \quad \text{(D.6a)}
$$

$$
\begin{pmatrix} 0 \\ \dot{\rho}'_S(t) \end{pmatrix} + \frac{1}{\kappa} \begin{pmatrix} \dot{\beta}'(t) \\ \dot{\gamma}'_S(t) \end{pmatrix} = -H_{(\beta,S),(\beta,S)} \begin{pmatrix} d_\beta(t) \\ d_{\gamma,S}(t) \end{pmatrix}, \quad \text{(D.6b)}
$$

$$
\rho'_S(t) \in \partial \|\gamma'_S(t)\|_1, \quad \text{(D.6c)}
$$

Define the *potential function* of the Oracle Dynamics (D.6) as

$$
\Psi(t) := D^{\rho'_S(t)}(\gamma_S^o, \gamma'_S(t)) + d(t)^2/(2\kappa),
$$

where $d(t)$ is defined in (D.5), and the Bregman distance

$$
D^{\rho'_S(t)}(\gamma_S^o, \gamma'_S(t)) := \|\gamma_S^o\|_1 - \|\gamma'_S(t)\|_1 - \langle\gamma_S^o - \gamma'_S(t), \, \rho'_S(t)\rangle = \|\gamma_S^o\|_1 - \langle\gamma_S^o, \rho'_S(t)\rangle.
$$

**Lemma 8** (Generalized Bihari's inequality)**.** *For all $t \geq 0$ we have*

$$\frac{\mathrm{d}}{\mathrm{d}t}\Psi(t) \leq -\lambda_H F^{-1}\left(\Psi(t)\right),$$

*where $\gamma_{\min}^o := \min(|\gamma_j^o| : \gamma_j^o \neq 0)$, and*

$$F(x) := \frac{x}{2\kappa} + \begin{cases} 0, & 0 \leq x < (\gamma_{\min}^o)^2, \\ 2x/\gamma_{\min}^o, & (\gamma_{\min}^o)^2 \leq x < s(\gamma_{\min}^o)^2, \\ 2\sqrt{sx}, & x \geq s(\gamma_{\min}^o)^2, \end{cases}$$

$$F^{-1}(x) := \inf(y: \ F(y) \geq x) \ (y \geq 0).$$

*Proof of Lemma 8.* Since

$$\begin{pmatrix} \beta'(t) \\ \gamma'(t) \end{pmatrix}, \ \begin{pmatrix} \beta^o \\ \gamma^o \end{pmatrix} \in L \oplus \mathbb{R}^s \oplus \{0\}^{m-s},$$

by (D.3) and Pythagorean Theorem,

$$\begin{aligned}
\ell\left(\beta'(t), \gamma'(t)\right) &= \frac{1}{2n}\left\| \begin{pmatrix} y \\ 0 \end{pmatrix} - \begin{pmatrix} X & 0 \\ -\sqrt{n/\nu}D & I_m \end{pmatrix}\begin{pmatrix} \beta'(t) \\ \gamma'(t) \end{pmatrix} \right\|_2^2 \\
&= \frac{1}{2n}\left\| \begin{pmatrix} X & 0 \\ -\sqrt{n/\nu}D & I_m \end{pmatrix}\begin{pmatrix} \beta'(t) \\ \gamma'(t) \end{pmatrix} - \begin{pmatrix} X & 0 \\ -\sqrt{n/\nu}D & I_m \end{pmatrix}\begin{pmatrix} \beta^o \\ \gamma^o \end{pmatrix} \right\|_2^2 \\
&\quad + \frac{1}{2n}\left\| \begin{pmatrix} y \\ 0 \end{pmatrix} - \begin{pmatrix} X & 0 \\ -\sqrt{n/\nu}D & I_m \end{pmatrix}\begin{pmatrix} \beta^o \\ \gamma^o \end{pmatrix} \right\|_2^2 \\
&\hspace{4cm} = L(t) + \text{constant (independent of } t), \quad \text{(D.7)}
\end{aligned}$$

where

$$\begin{aligned}
L(t) &:= \frac{1}{2n}\left\| \begin{pmatrix} X & 0 \\ -\sqrt{n/\nu}D & I_m \end{pmatrix}\begin{pmatrix} d_\beta(t) \\ d_\gamma(t) \end{pmatrix} \right\|_2^2 = \frac{1}{2}\left(d_\beta(t)^T, d_\gamma(t)^T\right) H \begin{pmatrix} d_\beta(t) \\ d_\gamma(t) \end{pmatrix} \\
&= \frac{1}{2}\left(d_\beta(t)^T, d_{\gamma,S}(t)^T\right) H_{(\beta,S),(\beta,S)} \begin{pmatrix} d_\beta(t) \\ d_{\gamma,S}(t) \end{pmatrix}. \quad \text{(D.8)}
\end{aligned}$$

Noting $\gamma_j(t) \cdot \dot{\rho}_j(t) \equiv 0$ for each $j$, by (D.6c) and (D.8) we have

$$\begin{aligned}
\frac{\mathrm{d}}{\mathrm{d}t}\Psi(t) &= \langle -\gamma_S^o, \dot{\rho}_S'(t)\rangle + d_{\gamma,S}(t)^T \dot{\gamma}_S(t)/\kappa + d_\beta(t)^T \dot{\beta}'(t)/\kappa \\
&= \left\langle \begin{pmatrix} d_\beta(t) \\ d_{\gamma,S}(t) \end{pmatrix}, \begin{pmatrix} 0 \\ \dot{\rho}_S'(t) \end{pmatrix} + \frac{1}{\kappa}\begin{pmatrix} \dot{\beta}'(t) \\ \dot{\gamma}_S'(t) \end{pmatrix} \right\rangle = -2L(t).
\end{aligned} \quad \text{(D.9)}$$

Thus it suffices to show

$$F\left(\frac{2}{\lambda_H}L(t)\right) \geq \Psi(t).$$

Since $\|\gamma_S^o\|_1 - \langle\gamma_S^o, \rho_S'(t)\rangle = 0$ if $\|\gamma_S'(t) - \gamma_S^o\|_2^2 < (\gamma_{\min}^o)^2$, and

$$\|\gamma_S^o\|_1 - \langle\gamma_S^o, \rho_S'(t)\rangle \leq 2\sum_{j \in N(t)}|\gamma_j^o| \ \left(N(t) := \{j: \ \text{sign}\left(\gamma_j'(t)\right) \neq \text{sign}\left(\gamma_j^o\right)\}\right)$$

$$\leq \begin{cases} \dfrac{2}{\gamma_{\min}^o}\displaystyle\sum_{j \in N(t)}(\gamma_j^o)^2 \leq \dfrac{2}{\gamma_{\min}^o}\|\gamma_S'(t) - \gamma_S^o\|_2^2 \\ 2\sqrt{s\displaystyle\sum_{j \in N(t)}(\gamma_j^o)^2} \leq 2\sqrt{s\|\gamma_S'(t) - \gamma_S^o\|_2^2}. \end{cases}$$

Thus

$$\Psi(t) - \frac{1}{2\kappa}\left(\|d_{\gamma,S}(t)\|_2^2 + \|d_\beta(t)\|_2^2\right) \leq F\left(\|d_{\gamma,S}(t)\|_2^2\right) - \frac{1}{2\kappa}\|d_{\gamma,S}(t)\|_2^2.$$

It suffice to show

$$F\left(\frac{2}{\lambda_H}L(t)\right) \geq F\left(\|d_{\gamma,S}(t)\|_2^2\right) + \frac{1}{2\kappa}\|d_\beta(t)\|_2^2,$$

which is true since by Assumption 1

$$2L(t) = \left(d_\beta(t)^T, d_{\gamma,S}(t)^T\right) \cdot H_{(\beta,S),(\beta,S)} \cdot \begin{pmatrix} d_\beta(t) \\ d_{\gamma,S}(t) \end{pmatrix} \geq \lambda_H \cdot d(t)^2, \tag{D.10}$$

and by $F(\cdot + x) \geq F(\cdot) + x/(2\kappa)$

$$F\left(d(t)^2\right) = F\left(\|d_\beta(t)\|_2^2 + \|d_{\gamma,S}(t)\|_2^2\right) \geq F\left(\|d_{\gamma,S}(t)\|_2^2\right) + \frac{1}{2\kappa}\|d_\beta(t)\|_2^2.$$

$\square$

**Proposition 3.** *Let* $\gamma_{\min}^o := \min(|\gamma_j^o| : \gamma_j^o \neq 0)$. *For*

$$t \geq \tau_\infty(\mu) := \frac{1}{\kappa\lambda_H}\log\frac{1}{\mu} + \frac{2\log s + 4 + d(0)/\kappa}{\lambda_H\gamma_{\min}^o} \quad (0 < \mu < 1), \tag{D.11}$$

*we have*

$$d(t) \leq \mu\gamma_{\min}^o \left(\Longrightarrow \operatorname{sign}(\gamma_S'(t)) = \operatorname{sign}(\gamma_S^o)\right), \text{ if } \gamma_j^o \neq 0 \text{ for } j \in S. \tag{D.12}$$

*For* $t \geq 0$, *we have*

$$d(t) \leq \min\left(\frac{4\sqrt{s} + d(0)/\kappa}{\lambda_H t}, \sqrt{\frac{2\left(1 + \nu\Lambda_X^2 + \Lambda_D^2\right)}{\lambda_H\nu}} \cdot d(0)\right). \tag{D.13}$$

*Proof of Proposition 3.* Noting (D.7) and that $\ell(\beta'(t), \gamma'(t))$ is *non-increasing*, we know $L(t)$ is *non-increasing*. (D.9) tells that $\Psi(t)$ is non-increasing since $L(t) \geq 0$. If $L(t) = 0$ for $t = \tau_\infty(\mu)$, by (D.10) and the fact that $L(t)$ is non-increasing, we have

$$d(t)^2 \leq \frac{2}{\lambda_H}L(t)^2 = 0 \quad (t \geq \tau_\infty(\mu)).$$

Therefore (D.12) holds for $t \geq \tau_\infty(\mu)$. Now assume that $L(t) > 0$ for $t = \tau_\infty(\mu)$ (and hence for $0 \leq t \leq \tau_\infty(\mu)$), then $\Psi(t)$ is *strictly* decreasing on $[0, \tau_\infty(\mu)]$. Besides, $F$ is strictly increasing and continuous on $[(\gamma_{\min}^o)^2, +\infty)$. Moreover,

$$F\left(d(0)^2\right) \geq F\left(\|\gamma_S^o\|_2^2\right) + \|\beta^o\|_2^2/(2\kappa) \geq \Psi(0),$$

$$d(0)^2 \geq \|\gamma_S^o\|_2^2 \geq s\left(\gamma_{\min}^o\right)^2,$$

If there does not exist some $t \leq \tau_\infty(\mu)$ satisfying (D.12), then for $0 \leq t \leq \tau_\infty(\mu)$,

$$\Psi(t) \begin{cases} \geq d(t)^2/(2\kappa) \geq \mu^2\left(\gamma_{\min}^o\right)^2/(2\kappa) > 0, & \text{if } \kappa < +\infty, \\ > 0, & \text{if } \kappa = +\infty, \end{cases}$$

which also implies that $F^{-1}(\Psi(t)) > 0$. By Lemma 8,

$$\lambda_H\tau_\infty(\mu) \leq \int_0^{\tau_\infty(\mu)}\frac{-\frac{d}{dt}\Psi(t)}{F^{-1}(\Psi(t))}dt = \int_{\Psi(\tau_\infty(\mu))}^{\Psi(0)}\frac{dx}{F^{-1}(x)}$$

$$\leq \left(\int_{\mu^2\left(\gamma_{\min}^o\right)^2/(2\kappa)}^{\left(\gamma_{\min}^o\right)^2/(2\kappa)} + \int_{\left(\gamma_{\min}^o\right)^2/(2\kappa)}^{F\left((\gamma_{\min}^o)^2\right)} + \int_{F\left((\gamma_{\min}^o)^2\right)}^{F\left(s(\gamma_{\min}^o)^2\right)} + \int_{F\left(s(\gamma_{\min}^o)^2\right)}^{F\left(d(0)^2\right)}\right)\frac{dx}{F^{-1}(x)}$$

$$\leq \int_{\mu^2\left(\gamma_{\min}^o\right)^2/(2\kappa)}^{\left(\gamma_{\min}^o\right)^2/(2\kappa)}\frac{dx}{2\kappa x} + \int_{\left(\gamma_{\min}^o\right)^2/(2\kappa)}^{F\left((\gamma_{\min}^o)^2\right)}\frac{1}{\left(\gamma_{\min}^o\right)^2}dx + \int_{\left(\gamma_{\min}^o\right)^2}^{s\left(\gamma_{\min}^o\right)^2}\frac{dF(x)}{x} + \int_{s\left(\gamma_{\min}^o\right)^2}^{d(0)^2}\frac{dF(x)}{x}$$

$$= \frac{1}{2\kappa}\log\frac{1}{\mu^2} + \frac{2}{\gamma_{\min}^o} + \int_{\left(\gamma_{\min}^o\right)^2}^{s\left(\gamma_{\min}^o\right)^2}\left(\frac{1}{2\kappa x} + \frac{2}{\gamma_{\min}^o x}\right)dx + \int_{s\left(\gamma_{\min}^o\right)^2}^{d(0)^2}\left(\frac{1}{2\kappa x} + \frac{\sqrt{s}}{x\sqrt{x}}\right)dx$$

$$< \frac{1}{2\kappa}\log\frac{1}{\mu^2} + \frac{2}{\gamma_{\min}^o} + \frac{1}{2\kappa}\log\frac{d(0)^2}{\left(\gamma_{\min}^o\right)^2} + \frac{2\log s}{\gamma_{\min}^o} + \frac{2}{\gamma_{\min}^o}$$

$$\leq \frac{1}{\kappa}\log\frac{1}{\mu} + \frac{2\log s + 4 + d(0)/\kappa}{\gamma_{\min}^o},$$

contradicting with the definition of $\tau_\infty(\mu)$. Thus (D.12) holds for some $0 \le \tau \le \tau_\infty(\mu)$. If $\kappa = +\infty$, we see that for $t \ge \tau_\infty(\mu)$, $\Psi(t) \le \Psi(\tau) = 0$. Then $-2L(t)$, the derivative of $\Psi(t)$, is $0$ (which means $d(t) = 0$) when $t \ge \tau_\infty(\mu)$, and (D.12) holds. If $\kappa < +\infty$, just note that for $t \ge \tau$,

$$d(t)^2/(2\kappa) \le \Psi(t) \le \Psi(\tau) = d(\tau)^2/(2\kappa) \implies d(t) \le d(\tau) \le \mu\gamma_{\min}^o.$$

So (D.12) holds for $t \ge \tau_\infty(\mu)$.

For any $t > 0$, if $L(t) = 0$, then $d(t) = 0$ and (D.13) holds. If $L(t) > 0$, let $C = \sqrt{2L(t)/\lambda_H} > 0$, then for any $0 \le t' \le t$,

$$\frac{\mathrm{d}}{\mathrm{d}t'}\Psi\left(t'\right) = -2L\left(t'\right) \le -2L(t) = -\lambda_H C^2.$$

Besides, for $\tilde{F}(x) = x/(2\kappa) + 2\sqrt{sx} \ge F(x)$, by Lemma 8 we have

$$\frac{\mathrm{d}}{\mathrm{d}t'}\Psi\left(t'\right) \le -\lambda_H F^{-1}\left(\Psi\left(t'\right)\right) \le -\lambda_H \tilde{F}^{-1}\left(\Psi\left(t'\right)\right).$$

By (D.9) and the fact that

$$\tilde{F}\left(d(0)^2\right) \ge \tilde{F}\left(\|\gamma_S^o\|_2^2\right) + \|\beta^o\|_2^2/(2\kappa) \ge \Psi(0),$$

we have that, if $d(0) > C$, then

$$\lambda_H t \le \int_0^t \frac{-\frac{\mathrm{d}}{\mathrm{d}t'}\Psi\left(t'\right)}{\max\left(C^2, \tilde{F}^{-1}\left(\Psi\left(t'\right)\right)\right)}\mathrm{d}t' = \int_{\Psi(t)}^{\Psi(0)} \frac{\mathrm{d}x}{\max\left(C^2, \tilde{F}^{-1}(x)\right)}$$

$$\le \int_{\tilde{F}(0)}^{\tilde{F}\left(d(0)^2\right)} \frac{\mathrm{d}x}{\max\left(C^2, \tilde{F}^{-1}(x)\right)} = \int_{\tilde{F}(0)}^{\tilde{F}(C^2)} \frac{\mathrm{d}x}{C^2} + \int_{C^2}^{d(0)^2} \frac{\mathrm{d}\tilde{F}(x)}{x}$$

$$= \frac{C^2/(2\kappa) + 2\sqrt{s}C}{C^2} + \int_{C^2}^{d(0)^2} \left(\frac{1}{2\kappa x} + \frac{\sqrt{s}}{x\sqrt{x}}\right)\mathrm{d}x$$

$$\le \frac{4\sqrt{s}}{C} + \frac{1}{2\kappa}\left(1 + \log\frac{d(0)^2}{C^2}\right) \le \frac{4\sqrt{s} + d(0)/\kappa}{C}.$$

If $d(0) \le C$, then similarly

$$\lambda_H t \le \int_{\tilde{F}(0)}^{\tilde{F}\left(d(0)^2\right)} \frac{\mathrm{d}x}{\max\left(C^2, \tilde{F}^{-1}(x)\right)} \le \int_{\tilde{F}(0)}^{\tilde{F}\left(d(0)^2\right)} \frac{\mathrm{d}x}{C^2}$$

$$= \frac{d(0)^2/(2\kappa) + 2\sqrt{s} \cdot d(0)}{C^2} \le \frac{4\sqrt{s} + d(0)/\kappa}{C}.$$

Combining it with (D.10), we have

$$d(t)^2 \le \frac{2}{\lambda_H}L(t) = \frac{2}{\lambda_H} \cdot \frac{\lambda_H C^2}{2} \le \left(\frac{4\sqrt{s} + M \cdot d(0)/\kappa}{\lambda_H t}\right)^2.$$

Besides, noting (C.8), we have

$$2L(0) = \left(d_\beta(0)^T, d_{\gamma,S}(0)^T\right) H_{(\beta,S),(\beta,S)} \begin{pmatrix} d_\beta(0) \\ d_{\gamma,S}(0) \end{pmatrix}$$

$$\le \|H\|_2 \cdot \left\|\begin{pmatrix} d_\beta(0) \\ d_\gamma(0) \end{pmatrix}\right\|_2^2 \le \frac{2\left(1 + \nu\Lambda_X^2 + \Lambda_D^2\right)}{\nu} \cdot d(0)^2.$$

Thus

$$d(t)^2 \le \frac{2}{\lambda_H}L(t) \le \frac{2}{\lambda_H}L(0) \le \frac{2\left(1 + \nu\Lambda_X^2 + \Lambda_D^2\right)}{\lambda_H\nu} \cdot d(0)^2.$$

Thus (D.13) holds. $\qquad\square$

## D.2 Oracle Iteration of Split LBI

Similarly, we define the *Oracle Iteration* of Split LBI as an *Oracle* version of Split LBI (C.2), with $S$ known and $\rho_{k,S^c}, \gamma_{k,S^c}$ set to be 0. Define

$$\Psi_k := \|\gamma_S^o\|_1 - \langle \gamma_S^o, \rho_{k,S} \rangle + \|\gamma_{k,S} - \gamma_S^o\|_2^2/(2\kappa) + \|\beta_k - \beta^o\|_2^2/(2\kappa).$$

Then we have

**Lemma 9** (Discrete Generalized Bihari's inequality). *Suppose $\kappa\alpha\|H\|_2 < 2$ and $\lambda_H' = \lambda_H(1 - \kappa\alpha\|H\|_2/2)$. For all $k$ we have*

$$\Psi_{k+1} - \Psi_k \leq -\alpha\lambda_H' F^{-1}(\Psi_k),$$

*where $\gamma_{\min}^o$, $F(x)$, $F^{-1}(x)$ are defined the same as in Lemma 8.*

*Proof of Lemma 9.* The proof is almost a discrete version of the continuous case. The only non-trivial thing is to show that

$$\Psi_{k+1} - \Psi_k \leq -2\alpha\left(1 - \kappa\alpha\|H\|_2/2\right)L_k, \text{ where}$$

$$L_k := \frac{1}{2}\left(d_{k,\beta}^T, d_{k,\gamma,S}^T\right) H_{(\beta,S),(\beta,S)} \begin{pmatrix} d_{k,\beta} \\ d_{k,\gamma,S} \end{pmatrix}, \begin{pmatrix} d_{k,\beta} \\ d_{k,\gamma,S} \end{pmatrix} := \begin{pmatrix} \beta_k' - \beta^o \\ \gamma_{k,S}' - \gamma_S^o \end{pmatrix}.$$

By (C.2), we have

$$-\alpha H_{(\beta,S),(\beta,S)} \begin{pmatrix} d_{k,\beta} \\ d_{\gamma,k,S} \end{pmatrix} = \begin{pmatrix} 0 \\ \rho_{k+1,S}' - \rho_{k,S}' \end{pmatrix} + \frac{1}{\kappa}\begin{pmatrix} \beta_{k+1}' - \beta_k' \\ \gamma_{k+1,S}' - \gamma_{k,S}' \end{pmatrix}.$$

Noting $(\rho_{k+1,S}' - \rho_{k,S}')^T \gamma_{k+1,S}' \geq 0$ and multiplying $(d_{k,\beta}^T, d_{\gamma,k,S}^T)$ on both sides, we have

$$-2\alpha L_k = d_{\gamma,k,S}^T \left(\rho_{k+1,S}' - \rho_{k,S}'\right) + \frac{1}{\kappa}\begin{pmatrix} d_{k,\beta} \\ d_{k,\gamma,S} \end{pmatrix}^T \begin{pmatrix} \beta_{k+1}' - \beta_k' \\ \gamma_{k+1,S}' - \gamma_{k,S}' \end{pmatrix}$$

$$\geq -\left(\rho_{k+1,S}' - \rho_{k,S}'\right)^T \left(\gamma_{k+1,S}' - \gamma_{k,S}'\right) - \left(\rho_{k+1,S}' - \rho_{k,S}'\right)^T \gamma_S^o$$

$$+ \frac{1}{\kappa}\begin{pmatrix} d_{k,\beta} \\ d_{k,\gamma,S} \end{pmatrix}^T \begin{pmatrix} \beta_{k+1}' - \beta_k' \\ \gamma_{k+1,S}' - \gamma_{k,S}' \end{pmatrix}.$$

Thus

$$\Psi_{k+1} - \Psi_k = -\left(\rho_{k+1,S}' - \rho_{S,k}'\right)^T \gamma_S^o + \frac{1}{2\kappa}\left(\left\|\begin{pmatrix} d_{k+1,\beta} \\ d_{k+1,\gamma,S} \end{pmatrix}\right\|_2^2 - \left\|\begin{pmatrix} d_{k,\beta} \\ d_{k,\gamma,S} \end{pmatrix}\right\|_2^2\right)$$

$$= -\left(\rho_{k+1,S}' - \rho_{S,k}'\right)^T \gamma_S^o + \frac{1}{2\kappa}\begin{pmatrix} \beta_{k+1}' - \beta_k' \\ \gamma_{k+1,S}' - \gamma_{k,S}' \end{pmatrix}^T \left(\begin{pmatrix} \beta_{k+1}' - \beta_k' \\ \gamma_{k+1,S}' - \gamma_{k,S}' \end{pmatrix} + 2\begin{pmatrix} d_{k,\beta} \\ d_{k,\gamma,S} \end{pmatrix}\right)$$

$$\leq -2\alpha L_k + \left(\rho_{S,k+1}' - \rho_{S,k}'\right)^T \left(\gamma_{k+1,S}' - \gamma_{k,S}'\right) + \frac{1}{2\kappa}\left\|\begin{pmatrix} \beta_{k+1}' - \beta_k' \\ \gamma_{k+1,S}' - \gamma_{k,S}' \end{pmatrix}\right\|_2^2$$

$$\leq -2\alpha L_k + \frac{\kappa}{2}\left\|\begin{pmatrix} 0 \\ \rho_{k+1,S}' - \rho_{k,S}' \end{pmatrix} + \frac{1}{\kappa}\begin{pmatrix} \beta_{k+1}' - \beta_k' \\ \gamma_{k+1,S}' - \gamma_{k,S}' \end{pmatrix}\right\|_2^2$$

$$= -\left(d_{k,\beta}^T, d_{k,\gamma,S}^T\right)\left(\alpha H_{(\beta,S),(\beta,S)} - \frac{\kappa\alpha^2}{2}H_{(\beta,S),(\beta,S)}^2\right)\begin{pmatrix} d_{k,\beta} \\ d_{k,\gamma,S} \end{pmatrix}$$

$$\leq -\alpha\left(1 - \frac{\kappa\alpha}{2}\left\|H_{(\beta,S),(\beta,S)}\right\|_2\right)\left(d_{k,\beta}^T, d_{k,\gamma,S}^T\right) H_{(\beta,S),(\beta,S)}\begin{pmatrix} d_{k,\beta} \\ d_{k,\gamma,S} \end{pmatrix}$$

$$\leq -2\alpha\left(1 - \kappa\alpha\|H\|_2/2\right)L_k.$$

$\square$

**Proposition 4.** *Suppose $\kappa\alpha\|H\|_2 < 2$ and $\lambda_H' = \lambda_H(1 - \kappa\alpha\|H\|_2/2)$. Let*

$$\gamma_{\min}^o := \min(|\gamma_j^o| : \gamma_j^o \neq 0),$$

$$d_{k,\beta} = \beta_k' - \beta^o, \ d_{k,\gamma} = \gamma_k' - \gamma^o, \ d_k = \sqrt{\|d_{k,\beta}\|_2^2 + \|d_{k,\gamma,S}\|_2^2}.$$

*Then for any $k$ such that*

$$k\alpha \geq \tau'_\infty(\mu) := \frac{1}{\kappa\lambda'_H}\log\frac{1}{\mu} + \frac{2\log s + 4 + d_0/\kappa}{\lambda'_H\gamma^o_{\min}} + 4\alpha \ (0 < \mu < 1), \tag{D.14}$$

*we have*

$$d_k \leq \mu\gamma^o_{\min}\left(\Longrightarrow \text{sign}\left(\gamma'_{k,S}\right) = \text{sign}\left(\gamma^o_S\right)\right), \text{ if } \gamma^o_j \neq 0 \text{ for } j \in S. \tag{D.15}$$

*For any $k$, we have*

$$d_k \leq \min\left(\frac{4\sqrt{s} + d_0/\kappa}{\lambda'_H k\alpha}, \sqrt{\frac{2\left(1 + \nu\Lambda_X^2 + \Lambda_D^2\right)}{\lambda'_H\nu}} \cdot d_0\right). \tag{D.16}$$

*Proof of Proposition 4.* The proof is almost a discrete version of the continuous case. The only non-trivial thing is described as follows. First, suppose there does not exist $k \leq \tau'_\infty(\mu)/\alpha$ satisfying (D.15), then for any $0 \leq \kappa\alpha \leq \tau'_\infty(\mu)$, we have $\Psi_k > \mu^2(\gamma^o_{\min})^2/(2\kappa)$. Letting $k_0 = 0$, then $\Psi_{k_0} = \Psi_0 \leq F(d_0^2)$. Suppose that

$$F\left(d_0^2\right) \geq \Psi_{k_0}, \ldots, \Psi_{k_1-1} > F\left(s\left(\gamma^o_{\min}\right)^2\right) \geq \Psi_{k_1}, \ldots, \Psi_{k_2-1} > F\left(\left(\gamma^o_{\min}\right)^2\right)$$

$$\geq \Psi_{k_2}, \ldots, \Psi_{k_3-1} > \left(\gamma^o_{\min}\right)^2/(2\kappa) \geq \Psi_{k_3}, \ldots, \Psi_{k_4-1} > \mu^2\left(\gamma^o_{\min}\right)^2/(2\kappa) \geq \Psi_{k_4}, \ldots$$

Then $k_4\alpha > \tau'_\infty(\mu)$. Besides, by Lemma 9,

$$\alpha \leq \frac{\Psi_k - \Psi_{k+1}}{\lambda'_H F^{-1}(\Psi_k)} \ (0 \leq k\alpha \leq \tau'_\infty(\mu)).$$

Thus $\lambda'_H(k_4 - 4)\alpha$ is not greater than

$$\left(\sum_{k=k_3}^{k_4-2} + \sum_{k=k_2}^{k_3-2} + \sum_{k=k_1}^{k_2-2} + \sum_{k=k_0}^{k_1-2}\right)\frac{\Psi_k - \Psi_{k+1}}{F^{-1}(\Psi_k)} \leq \sum_{k=k_3}^{k_4-2}\frac{\Psi_k - \Psi_{k+1}}{2\kappa\Psi_k} + \sum_{k=k_2}^{k_3-2}\frac{\Psi_k - \Psi_{k+1}}{\left(\gamma^o_{\min}\right)^2}$$

$$+ \sum_{k=k_1}^{k_2-2}\frac{F(\Delta_k) - F(\Delta_{k+1})}{\Delta_k} + \sum_{k=k_0}^{k_1-2}\frac{F(\Delta_k) - F(\Delta_{k+1})}{\Delta_k} \ \left(\Delta_k := F^{-1}(\Psi_k)\right)$$

$$= \sum_{k=k_3}^{k_4-2}\frac{\Psi_k - \Psi_{k+1}}{2\kappa\Psi_k} + \sum_{k=k_2}^{k_3-2}\frac{\Psi_k - \Psi_{k+1}}{\left(\gamma^o_{\min}\right)^2} + \sum_{k=k_1}^{k_2-2}\left(\frac{\Delta_k - \Delta_{k+1}}{2\kappa\Delta_k} + \frac{2(\Delta_k - \Delta_{k+1})}{\gamma^o_{\min}\Delta_k}\right)$$

$$+ \sum_{k=k_0}^{k_1-2}\left(\frac{\Delta_k - \Delta_{k+1}}{2\kappa\Delta_k} + \frac{2\sqrt{s}\left(\sqrt{\Delta_k} - \sqrt{\Delta_{k+1}}\right)}{\Delta_k}\right).$$

By $(u - v)/u \leq \log(u/v)$ and $(\sqrt{u} - \sqrt{v})/u \leq 1/\sqrt{v} - 1/\sqrt{u}$ for $u \geq v > 0$, the quantity above is not greater than

$$\frac{\log\left(\Psi_{k_3}/\Psi_{k_4-1}\right)}{2\kappa} + \frac{\Psi_{k_2} - \Psi_{k_3-1}}{\left(\gamma^o_{\min}\right)^2}$$

$$+ \frac{\log\left(\Delta_{k_0}/\Delta_{k_2-1}\right)}{2\kappa} + \frac{2\log\left(\Delta_{k_1}/\Delta_{k_2-1}\right)}{\gamma^o_{\min}} + 2\sqrt{s}\left(\frac{1}{\sqrt{\Delta_{k_1-1}}} - \frac{1}{\sqrt{\Delta_{k_0}}}\right)$$

$$< \frac{\log\left(1/\mu^2\right)}{2\kappa} + \frac{2\gamma^o_{\min}}{\left(\gamma^o_{\min}\right)^2} + \frac{\log\left(d_0^2/\left(\gamma^o_{\min}\right)^2\right)}{2\kappa} + \frac{2\log s}{\gamma^o_{\min}} + \frac{2\sqrt{s}}{\sqrt{s\left(\gamma^o_{\min}\right)^2}}.$$

Therefore we get

$$\lambda'_H\left(\tau'_\infty(\mu) - 4\alpha\right) < \lambda'_H\left(k_4 - 4\right)\alpha < \frac{1}{\kappa}\log\frac{1}{\mu} + \frac{2\log s + 4 + d_0/\kappa}{\gamma^o_{\min}},$$

a contradiction with the definition of $\tau'_\infty(\mu)$. So there exists some $k \leq \tau'_\infty(\mu)/\alpha$ satisfying (D.15). Then continue to imitate the proof in the continous version, we obtain (D.15) for all $t \geq \tau'_\infty(\mu)$. The proof of (D.16) follows the same spirit. □

# E Proofs of Consistency of Split LBI

*Proof of Theorem 2 and 3.* They are merely discrete versions of proofs of the following Theorem 4 and 5, but applying Lemma 9 and Proposition 4 instead of Lemma 8 and Proposition 3. □

**Theorem 4** (Consistency of Split (LB)ISS). *Under Assumption 1 and 2, suppose $\kappa$ is large enough to satisfy* (2.12). *Let*

$$\overline{\tau} := \frac{\eta}{8\sigma} \cdot \frac{\lambda_D}{\Lambda_X} \sqrt{\frac{n}{\log m}}. \tag{E.1}$$

*Then with probability not less than $1 - 6/m - 3\exp(-4n/5)$, we have all the following properties.*

1. No-false-positive*: The solution has no false-positive, i.e. $\operatorname{supp}(\gamma(t)) \subseteq S$, for $0 \le t \le \overline{\tau}$.*

2. Sign consistency of $\gamma(t)$*: Once the $\gamma^\star_{\min}$ condition*

$$\gamma^\star_{\min} := (D_S \beta^\star)_{\min} \ge \frac{16\sigma}{\eta \lambda_H} \cdot \frac{\Lambda_X \Lambda_D}{\lambda_D^2} \left(2\log s + 5 + \log(8\Lambda_D)\right) \sqrt{\frac{\log m}{n}} \tag{E.2}$$

*holds, then $\gamma(t)$ has sign consistency at $\overline{\tau}$, i.e.*

$$\operatorname{sign}\left(\gamma\left(\overline{\tau}\right)\right) = \operatorname{sign}\left(D\beta^\star\right),$$

3. $\ell_2$ consistency of $\gamma(t)$*: For $0 \le t \le \overline{\tau}$,*

$$\|\gamma(t) - D\beta^\star\|_2 \le \frac{5\sqrt{s}}{\lambda_H t} + \frac{2\sigma}{\lambda_H} \cdot \frac{\Lambda_X}{\lambda_D} \sqrt{\frac{s\log m}{n}},$$

*Consequently,*

$$\|\gamma\left(\overline{\tau}\right) - D\beta^\star\|_2 \le \frac{42\sigma}{\eta \lambda_H} \cdot \frac{\Lambda_X}{\lambda_D} \sqrt{\frac{s\log m}{n}}.$$

4. $\ell_2$ consistency of $\beta(t)$*: For $0 \le t \le \overline{\tau}$,*

$$\|\beta(t) - \beta^\star\|_2 \le \frac{5\sqrt{s}}{\lambda_H t} + \frac{2\sigma}{\lambda_H} \cdot \frac{\lambda_1 \Lambda_X + \Lambda_X^2}{\lambda_1 \lambda_D^2} \sqrt{\frac{s\log m}{n}} + \frac{2\sigma}{\lambda_1} \sqrt{\frac{r'\log m}{n}}$$
$$+ \nu \cdot 2\sigma \cdot \frac{\lambda_1 \Lambda_X + \Lambda_X^2}{\lambda_1 \lambda_D^2}.$$

*Consequently,*

$$\|\beta\left(\overline{\tau}\right) - \beta^\star\|_2 \le \frac{42\sigma}{\eta \lambda_H} \cdot \frac{\lambda_1 \Lambda_X(1 + \lambda_D) + \Lambda_X^2}{\lambda_1 \lambda_D^2} \sqrt{\frac{s\log m}{n}} + \frac{2\sigma}{\lambda_1} \sqrt{\frac{r'\log m}{n}}$$
$$+ \nu \cdot 2\sigma \cdot \frac{\lambda_1 \Lambda_X + \Lambda_X^2}{\lambda_1 \lambda_D^2}.$$

**Theorem 5** (Consistency of revised version of Split LBISS). *Under Assumption 1 and 2, suppose $\kappa$ is large enough to satisfy* (2.12), *and $\overline{\tau}$ is defined the same as in Theorem 4. Define*

$$S(t) := \operatorname{supp}(\gamma(t)), \ P_{S(t)} := P_{\ker\left(D_{S(t)^c}\right)} = I - D^\dagger_{S(t)^c} D_{S(t)^c}, \ \tilde{\beta}(t) := P_{S(t)}\beta(t).$$

*If $S(t)^c = \varnothing$, define $P_{S(t)} = I$. Then we have the following properties.*

1. Sign consistency of $\tilde{\beta}(t)$*: If the $\gamma^\star_{\min}$ condition* (E.2) *holds, then with probability not less than $1 - 8/m - 3\exp(-4n/5)$,*

$$\operatorname{sign}\left(D\tilde{\beta}\left(\overline{\tau}\right)\right) = \operatorname{sign}\left(D\beta^\star\right).$$

2. $\ell_2$ consistency of $\tilde{\beta}(t)$: With probability not less than $1 - 8/m - 2r'/m^2 - 3\exp(-4n/5)$, we have that for $0 \le t \le \bar{\tau}$,

$$\left\|\tilde{\beta}(t) - \beta^\star\right\|_2 \le \left(\frac{10\sqrt{s}}{\lambda_H t} + \frac{2\sigma}{\lambda_H} \cdot \frac{\Lambda_X \Lambda_D}{\lambda_D^3}\sqrt{\frac{s\log m}{n}}\right)$$

$$+ \frac{2\sigma}{\lambda_H}\left(\frac{\Lambda_X}{\lambda_D^2} + \frac{\lambda_H \lambda_D^2 + \Lambda_X^2}{\lambda_1 \lambda_D^2}\right)\sqrt{\frac{r'\log m}{n}} + 2\left\|D^\dagger_{S(t)^c}D_{S(t)^c\cap S}\beta^\star\right\|_2.$$

Consequently, if additionally $S(\bar{\tau}) = S$, then the last term on the right hand side drops for $t = \bar{\tau}$, and we can easily obtain

$$\left\|\tilde{\beta}(\bar{\tau}) - \beta^\star\right\|_2 \le \frac{80\sigma}{\eta\lambda_H} \cdot \frac{\Lambda_X\left(\Lambda_D + \lambda_D^2\right)}{\lambda_D^3}\sqrt{\frac{s\log m}{n}}$$

$$+ \frac{2\sigma}{\lambda_H}\left(\frac{\Lambda_X}{\lambda_D^2} + \frac{\lambda_H\lambda_D^2 + \Lambda_X^2}{\lambda_1\lambda_D^2}\right)\sqrt{\frac{r'\log m}{n}}.$$

Before proving Theorem 4 and 5, we need the following lemmas.

**Lemma 10.** *Suppose* $\Sigma_{S,S} \succeq \lambda_\Sigma I$. *For* $\beta^o \in L$ *and* $\gamma^o_S \in \mathbb{R}^s$ *satisfying* (D.2), *we have*

$$\|\beta^o - \beta^\star\|_2^2 = \|\delta^o - \delta^\star\|_2^2 + \|\xi^o - \xi^\star\|_2^2, \text{ where}$$
$$\delta^o - \delta^\star := V^T(\beta^o - \beta^\star), \ \xi^o - \xi^\star = V_1^T\tilde{V}^T(\beta^o - \beta^\star), \tag{E.3}$$

*and*

$$\delta^o - \delta^\star = \underbrace{\left(\nu B^{-1} + B^{-1}\Lambda U_S^T\Sigma^{-1}_{S,S}U_S\Lambda B^{-1}\right)V^TX^*\left(I - U_1U_1^T\right)}_{\triangleq B_\delta}\epsilon, \text{ with } \|B_\delta\|_2 \le \frac{\Lambda_X}{\sqrt{n}\cdot\lambda_\Sigma\lambda_D^2}, \tag{E.4}$$

$$\xi^o - \xi^\star = \underbrace{n^{-1/2}\Lambda_1^{-1}U_1^T(I - XVB_\delta)}_{\triangleq B_\xi}\epsilon, \text{ with } \|B_\xi\|_2 \le \frac{\lambda_\Sigma\lambda_D^2 + \Lambda_X^2}{\sqrt{n}\cdot\lambda_1\lambda_\Sigma\lambda_D^2}. \tag{E.5}$$

*Besides, we have*

$$\gamma^o_S - \gamma^\star_S = \underbrace{\Sigma^{-1}_{S,S}U_S\Lambda B^{-1}V^TX^*\left(I - U_1U_1^T\right)}_{\triangleq B_\gamma}\epsilon, \text{ with } \|B_\gamma\|_2 \le \frac{\Lambda_X}{\sqrt{n}\cdot\lambda_\Sigma\lambda_D}. \tag{E.6}$$

*Proof.* By Lemma 3 and $\beta^o - \beta^\star \in L$, we have (E.3). By (D.2), we have

$$\gamma^o_S - \gamma^\star_S = D_S(\beta^o - \beta^\star) = U_S\Lambda(\delta^o - \delta^\star), \tag{E.7}$$

and

$$X^*\epsilon + D_S^T(\gamma^o_S - \gamma^\star_S)/\nu = \left(X^*X + D^TD/\nu\right)(\beta^o - \beta^\star),$$

i.e.

$$X^*\epsilon + V\Lambda U_S^T(\gamma^o_S - \gamma^\star_S)/\nu = \left(X^*X + V\Lambda^2V^T/\nu\right)\left(V(\delta^o - \delta^\star) + \tilde{V}V_1(\xi^o - \xi^\star)\right)$$
$$= \left(X^*XV + V\Lambda^2/\nu\right)(\delta^o - \delta^\star) + \sqrt{n}X^*U_1\Lambda_1(\xi^o - \xi^\star). \tag{E.8}$$

Left multiplying $\Lambda_1^{-2}V_1^T\tilde{V}^T$ on both sides of (E.8) leads to

$$\xi^o - \xi^\star = \frac{1}{\sqrt{n}}\Lambda_1^{-1}U_1^T\left(\epsilon - XV(\delta^o - \delta^\star)\right). \tag{E.9}$$

Then left multiplying $V^T$ on both sides of (E.8) leads to

$$V^TX^*\epsilon + \Lambda U_S^T(\gamma^o_S - \gamma^\star_S)/\nu$$
$$= \left(V^TX^*XV + \Lambda^2/\nu\right)(\delta^o - \delta^\star) + \sqrt{n}V^TX^*U_1\Lambda_1 \cdot \frac{1}{\sqrt{n}}\Lambda_1^{-1}U_1^T\left(\epsilon - XV(\delta^o - \delta^\star)\right)$$
$$= \left(V^TX^*\left(I - U_1U_1^T\right)XV + \Lambda^2/\nu\right)(\delta^o - \delta^\star) + V^TX^*U_1U_1^T\epsilon.$$

Recalling the definition of $B$ in Lemma 4, the equation above implies

$$\delta^o - \delta^\star = B^{-1} \Lambda U_S^T (\gamma_S^o - \gamma_S^\star) + \nu B^{-1} V^T X^* \left( I - U_1 U_1^T \right) \epsilon. \tag{E.10}$$

Plugging it into (E.7), we obtain $\gamma_S^o - \gamma_S^\star = B_\gamma \epsilon$. Then noting $B \succeq \lambda_D^2 I$, we have

$$\|B_\gamma\|_2 \leq \left\| \Sigma_{S,S}^{-1} \right\|_2 \cdot 1 \cdot \left\| \Lambda B^{-1} \right\|_2 \cdot 1 \cdot \|X^*\|_2 \cdot \left\| I - U_1 U_1^T \right\|_2 \leq \frac{\Lambda_X}{\sqrt{n} \cdot \lambda_\Sigma \lambda_D}.$$

so (E.6) holds. Now by (E.10) we have $\delta^o - \delta^\star = B_\delta \epsilon$. Noting (A.8) and $\Sigma_{S,S} \succeq \lambda_\Sigma I$, we have

$$U_S \Lambda B^{-1/2} \cdot B^{-1/2} \Lambda U_S^T \preceq (1 - \lambda_\Sigma \nu) I$$
$$\iff B^{-1/2} \Lambda U_S^T \cdot U_S \Lambda B^{-1/2} \preceq (1 - \lambda_\Sigma \nu) I \iff \Lambda U_S^T U_S \Lambda \preceq (1 - \lambda_\Sigma \nu) B.$$

Thus

$$\nu B^{-1} + B^{-1} \Lambda U_S^T \Sigma_{S,S}^{-1} U_S \Lambda B^{-1} \preceq \nu B^{-1} + \frac{1}{\lambda_\Sigma} B^{-1} \Lambda U_S^T U_S \Lambda B^{-1} \preceq \frac{1}{\lambda_\Sigma} B^{-1},$$

which immediately leads to (E.4). Finally, combining (E.9) with (E.4) we have (E.5). $\qquad\square$

**Lemma 11** (No-false-positive condition for Split LBISS). *For the Oracle Dynamics (D.6), if there is $\tau > 0$, such that for $0 \leq t \leq \tau$ the inequality*

$$\left\| H_{S^c,(\beta,S)} H_{(\beta,S),(\beta,S)}^\dagger \left( \begin{pmatrix} 0_p \\ \rho_S'(t) \end{pmatrix} + \frac{1}{\kappa} \begin{pmatrix} \beta'(t) \\ \gamma_S'(t) \end{pmatrix} - t \begin{pmatrix} X^*\epsilon \\ 0_s \end{pmatrix} \right) \right\|_\infty < 1 \tag{E.11}$$

*holds, then the solution path of the original dynamics (C.3) has no false-positive for $0 \leq t \leq \tau$.*

*Proof of Lemma 11.* It is easy to see that

$$\begin{pmatrix} 0_p \\ \dot\rho(t) \end{pmatrix} + \frac{1}{\kappa} \begin{pmatrix} \dot\beta(t) \\ \dot\gamma(t) \end{pmatrix} = H \left( \begin{pmatrix} \beta(t) \\ \gamma(t) \end{pmatrix} - \begin{pmatrix} \beta^\star \\ \gamma^\star \end{pmatrix} \right) + \begin{pmatrix} X^*\epsilon \\ 0_m \end{pmatrix}. \tag{E.12}$$

Now let

$$\bar\tau := \inf \left( t \geq 0 : \ \|\rho_{S^c}(t)\|_\infty = 1 \right).$$

It suffices to show $\bar\tau > \tau$. For $0 \leq t < \bar\tau$, we have $\gamma_{S^c}(t) = 0$, which also implies $\rho_S(t) = \rho_S'(t)$ and $\gamma_S(t) = \gamma_S'(t)$. Hence by (E.12) we have

$$\begin{pmatrix} 0_p \\ \dot\rho_S'(t) \end{pmatrix} + \frac{1}{\kappa} \begin{pmatrix} \dot\beta'(t) \\ \dot\gamma_S'(t) \end{pmatrix} = -H_{(\beta,S),(\beta,S)} \left( \begin{pmatrix} \beta'(t) \\ \gamma_S'(t) \end{pmatrix} - \begin{pmatrix} \beta^\star \\ \gamma_S^\star \end{pmatrix} \right) + \begin{pmatrix} X^*\epsilon \\ 0_s \end{pmatrix}, \tag{E.13}$$

$$\dot\rho_{S^c}(t) = -H_{S^c,(\beta,S)} \left( \begin{pmatrix} \beta'(t) \\ \gamma_S'(t) \end{pmatrix} - \begin{pmatrix} \beta^\star \\ \gamma_S^\star \end{pmatrix} \right).$$

We claim that

$$\begin{pmatrix} \beta'(t) \\ \gamma_S'(t) \end{pmatrix} - \begin{pmatrix} \beta^\star \\ \gamma_S^\star \end{pmatrix} \in L \oplus \mathbb{R}^s = \text{Im} \left( H_{(\beta,S),(\beta,S)}^\dagger \right)$$

(the equality above will be shown at last), so by (E.13) we have

$$\begin{pmatrix} \beta'(t) \\ \gamma_S'(t) \end{pmatrix} - \begin{pmatrix} \beta^\star \\ \gamma_S^\star \end{pmatrix} = -H_{(\beta,S),(\beta,S)}^\dagger \left( \begin{pmatrix} 0_p \\ \dot\rho_S'(t) \end{pmatrix} + \frac{1}{\kappa} \begin{pmatrix} \dot\beta'(t) \\ \dot\gamma_S'(t) \end{pmatrix} - \begin{pmatrix} X^*\epsilon \\ 0_s \end{pmatrix} \right),$$

$$\implies \dot\rho_{S^c}(t) = H_{S^c,(\beta,S)} H_{(\beta,S),(\beta,S)}^\dagger \left( \begin{pmatrix} 0_p \\ \dot\rho_S'(t) \end{pmatrix} + \frac{1}{\kappa} \begin{pmatrix} \dot\beta'(t) \\ \dot\gamma_S'(t) \end{pmatrix} - \begin{pmatrix} X^*\epsilon \\ 0_s \end{pmatrix} \right).$$

Integration on both sides leads to

$$\rho_{S^c}(t) = H_{S^c,(\beta,S)} H_{(\beta,S),(\beta,S)}^\dagger \left( \begin{pmatrix} 0_p \\ \rho_S'(t) \end{pmatrix} + \frac{1}{\kappa} \begin{pmatrix} \beta'(t) \\ \gamma_S'(t) \end{pmatrix} - t \begin{pmatrix} X^*\epsilon \\ 0_s \end{pmatrix} \right) \quad (0 \leq t < \bar\tau).$$

Due to the continuity of $\rho_{S^c}(t)$, $\rho_S'(t)$ (and $\gamma_S'(t)$, if $\kappa < +\infty$), the equation above also holds for $t = \bar\tau$. According to the definition of $\bar\tau$, we know (E.11) does not hold for $t = \bar\tau$. Thus $\bar\tau > \tau$, and the desired result follows.

So it suffices to prove

$$L \oplus \mathbb{R}^s = \mathrm{Im}\left(H^\dagger_{(\beta,S),(\beta,S)}\right). \tag{E.14}$$

Actually, let $H_{(\beta,S),(\beta,S)} = U'\Lambda'U'^T$ where $U'^TU' = I$ and $\Lambda'$ is an invertible diagonal matrix. It suffices to show $L \oplus \mathbb{R}^s = \mathrm{Im}(U')$. First, by the definition of $H$, one can easily verify that

$$\mathrm{Im}\left(U'\right) = \mathrm{Im}\left(H_{(\beta,S),(\beta,S)}\right) \subseteq \left(\mathrm{Im}\left(X^T\right) + \mathrm{Im}\left(D^T\right)\right) \oplus \mathbb{R}^s = L \oplus \mathbb{R}^s.$$

On the other hand, assume that $(U', \tilde{U}')$ is an orthogonal square matrix. For any $\zeta \in L \oplus \mathbb{R}^s$, since $P_{\mathrm{Im}(U')}\zeta \in \mathrm{Im}(U') \subseteq L \oplus \mathbb{R}^s$, we have $P_{\mathrm{Im}(\tilde{U}')}\zeta = \zeta - P_{\mathrm{Im}(U')}\zeta \in L \oplus \mathbb{R}^s$, and (2.2) tells us

$$0 = \left\|\Lambda'^{1/2}U'^T P_{\mathrm{Im}(\tilde{U}')}\zeta\right\|_2^2 \geq \lambda_H \left\|P_{\mathrm{Im}(\tilde{U}')}\zeta\right\|_2^2 \implies P_{\mathrm{Im}(\tilde{U}')}\zeta = 0$$
$$\implies \zeta = P_{\mathrm{Im}(U')}\zeta + P_{\mathrm{Im}(\tilde{U}')}\zeta = P_{\mathrm{Im}(U')}\zeta \in \mathrm{Im}(U').$$

Thus (E.14) holds. $\qquad\square$

*Proof of Theorem 4.* By Lemma 6, (A.3), (E.5) and (E.6), we have that with probability not less than $1 - 4s/m^2 \geq 1 - 4/m$,

$$\|\gamma_S^o - \gamma_S^\star\|_\infty < \frac{2\sigma}{\lambda_H} \cdot \frac{\Lambda_X}{\lambda_D} \sqrt{\frac{\log m}{n}}, \tag{E.15}$$

$$\|\xi^o - \xi^\star\|_\infty < \frac{2\sigma}{\lambda_H} \cdot \frac{\lambda_H \lambda_D^2 + \Lambda_X^2}{\lambda_1 \lambda_D^2} \sqrt{\frac{\log m}{n}}. \tag{E.16}$$

By (A.4) and (E.3) to (E.6), with probability not less than $1 - 3\exp(-4n/5)$,

$$\|\epsilon\|_2 \leq 2\sigma\sqrt{n}, \text{ which implies}$$

$$\|\gamma_S^o - \gamma_S^\star\|_2 < \frac{2\sigma}{\lambda_H} \cdot \frac{\Lambda_X}{\lambda_D}, \ \|\delta^o - \delta^\star\|_2 < \frac{2\sigma}{\lambda_H} \cdot \frac{\Lambda_X}{\lambda_D^2}, \ \|\xi^o - \xi^\star\|_2 < \frac{2\sigma}{\lambda_H} \cdot \frac{\lambda_H \lambda_D^2 + \Lambda_X^2}{\lambda_1 \lambda_D^2}. \tag{E.17}$$

The inequalities above also imply

$$\|\beta^o - \beta^\star\|_2 \leq \|\delta^o - \delta^\star\|_2 + \|\xi^o - \xi^\star\|_2 < \frac{2\sigma}{\lambda_H}\left(\frac{\Lambda_X}{\lambda_D^2} + \frac{\lambda_H \lambda_D^2 + \Lambda_X^2}{\lambda_1 \lambda_D^2}\right), \tag{E.18}$$

and

$$d(0) = \sqrt{\|\gamma_S^o\|_2^2 + \|\beta^o\|_2^2} \leq \|\gamma_S^\star\|_2 + \|\beta^\star\|_2 + \|\gamma_S^o - \gamma_S^\star\|_2 + \|\beta^o - \beta^\star\|_2$$
$$< (1 + \Lambda_D)\|\beta^\star\|_2 + \frac{2\sigma}{\lambda_H}\left(\frac{\Lambda_X}{\lambda_D} + \frac{\Lambda_X}{\lambda_D^2} + \frac{\lambda_H \lambda_D^2 + \Lambda_X^2}{\lambda_1 \lambda_D^2}\right). \tag{E.19}$$

From now, we assume all the inequalities above hold. The condition on $\kappa$ now tells us

$$\kappa \geq \frac{4}{\eta}\left(1 + \frac{1}{\lambda_D} + \frac{\Lambda_X}{\lambda_1 \lambda_D}\right)\left(1 + \sqrt{\frac{2\left(1 + \nu\Lambda_X^2 + \Lambda_D^2\right)}{\lambda_H \nu}}\right) \cdot d(0) \ (\geq d(0)). \tag{E.20}$$

Now we prove the *No-false-positive* property. By Lemma 11, it suffices to show that for $0 \le t \le \bar{\tau}$, (E.11) holds with probability not less that $1 - 2/m$. By (A.7), (A.15) and (D.13),

$$
\frac{1}{\kappa} \left\| H_{S^c,(\beta,S)} H^\dagger_{(\beta,S),(\beta,S)} \begin{pmatrix} \beta'(t) \\ \gamma'_S(t) \end{pmatrix} \right\|_\infty
$$

$$
= \left\| \left( -D_{S^c} A^\dagger + \Sigma_{S^c,S} \Sigma^{-1}_{S,S} D_S \right) A^\dagger \beta'(t) + \Sigma_{S^c,S} \Sigma^{-1}_{S,S} \gamma'_S(t) \right\|_\infty / \kappa
$$

$$
\le \left\| D_{S^c} A^\dagger \beta'(t) \right\|_\infty / \kappa + \left\| \Sigma_{S^c,S} \Sigma^{-1}_{S,S} D_S A^\dagger \beta'(t) \right\|_\infty / \kappa + \left\| \gamma'_S(t) \right\|_\infty / \kappa
$$

$$
\le 2 \left\| DA^\dagger \right\|_2 \cdot \left\| \beta'(t) \right\|_2 / \kappa + \left\| \gamma'_S(t) \right\|_2 / \kappa \le \left( 2 \left( \frac{1}{\lambda_D} + \frac{\Lambda_X}{\lambda_D \lambda_1} \right) + 1 \right) \sqrt{ \left\| \beta'(t) \right\|_2^2 + \left\| \gamma'_S(t) \right\|_2^2 } / \kappa
$$

$$
\le 2 \left( 1 + \frac{1}{\lambda_D} + \frac{\Lambda_X}{\lambda_D \lambda_1} \right) (d(0) + d(t)) / \kappa
$$

$$
\le 2 \left( 1 + \frac{1}{\lambda_D} + \frac{\Lambda_X}{\lambda_D \lambda_1} \right) \left( 1 + \sqrt{ \frac{2 \left( 1 + \nu \Lambda_X^2 + \Lambda_D^2 \right)}{\lambda_H \nu} } \right) d(0)/\kappa \le \frac{\eta}{2}.
$$

Besides, by (A.15) we have

$$
\left\| H_{S^c,(\beta,S)} H^\dagger_{(\beta,S),(\beta,S)} \begin{pmatrix} X^* \epsilon \\ 0 \end{pmatrix} \right\|_\infty = \left\| \left( -D_{S^c} + \Sigma_{S^c,S} \Sigma^\dagger_{S,S} D_S \right) A^\dagger X^* \epsilon \right\|_\infty
$$

$$
\le \left\| D_{S^c} A^\dagger X^* \epsilon \right\|_\infty + \left\| D_S A^\dagger X^* \epsilon \right\|_\infty \le 2 \left\| DA^\dagger X^* \epsilon \right\|_\infty.
$$

By (A.8), $DA^\dagger D^T = U \Lambda B^{-1} \Lambda U^T$ and $\Lambda^2 \preceq B \preceq (1 + \nu \Lambda_X^2 / \lambda_D^2) \Lambda^2$, therefore 1 is an upper bound of the largest eigenvalue of $DA^\dagger D^T$, and $1/(1 + \nu \Lambda_X^2 / \lambda_D^2)$ is a lower bound of the smallest *nonzero* eigenvalue of $DA^\dagger D^T$. Then

$$
DA^\dagger X^* \left( DA^\dagger X^* \right)^T = \frac{1}{n\nu} DA^\dagger \left( A - D^T D \right) A^\dagger D^T
$$

$$
= \frac{1}{n\nu} \left( DA^\dagger D^T - \left( DA^\dagger D^T \right)^2 \right) \preceq \frac{1}{n\nu} \min \left( \frac{1}{4}, \frac{\nu \Lambda_X^2 / \lambda_D^2}{\left( 1 + \nu \Lambda_X^2 / \lambda_D^2 \right)^2} \right) I \preceq \frac{\Lambda_X^2}{n \cdot \lambda_D^2} I.
$$

By (A.3), with probability not less than $1 - 2/m$, for any $0 \le t \le \bar{\tau}$,

$$
\left\| H_{S^c,(\beta,S)} H^\dagger_{(\beta,S),(\beta,S)} \cdot t \begin{pmatrix} X^* \epsilon \\ 0 \end{pmatrix} \right\|_\infty \le 2\bar{\tau} \left\| DA^\dagger X^* \epsilon \right\|_\infty \le 2\bar{\tau} \cdot 2\sigma \cdot \sqrt{ \frac{\Lambda_X^2}{n \cdot \lambda_D^2} } \cdot \sqrt{\log m} < \frac{\eta}{2}.
$$

Combining the results above with Assumption 2, we have for $0 \le t \le \bar{\tau}$, (E.11) holds with probability not less that $1 - 2/m$, and we have the No-false-positive property (which tells that $(\beta(t), \gamma_S(t))$ coincides with that of the Oracle Dynamics for $0 \le t \le \bar{\tau}$).

Then we prove the *sign consistency of $\gamma(t)$*. If the $\gamma^\star_{\min}$ condition (E.2) holds, by (E.15),

$$
\left\| \gamma_S^o - \gamma_S^\star \right\|_\infty \le \frac{2\sigma}{\lambda_H} \cdot \frac{\Lambda_X}{\lambda_D} \sqrt{\frac{\log m}{n}} \le \frac{\gamma^\star_{\min}}{2} \implies \gamma^o_{\min} \ge \frac{1}{2} \gamma^\star_{\min}. \tag{E.21}
$$

Thus $\text{sign}(\gamma_S^o) = \text{sign}(\gamma_S^\star)$, and

$$
\gamma^o_{\min} \ge \frac{1}{2} \gamma^\star_{\min} \ge \frac{2 \log s + 5}{\lambda_H \bar{\tau}} > \frac{2 \log s + 4 + d(0)/\kappa}{\lambda_H \bar{\tau}} \implies \bar{\tau} > \frac{2 \log s + 4 + d(0)/\kappa}{\lambda_H \gamma^o_{\min}}.
$$

By (D.12), the sign consistency of $\gamma'_S(t)$ holds for

$$
t > \inf_{0 < \mu < 1} \left( \frac{1}{\kappa \lambda_H} \log \frac{1}{\mu} + \frac{2 \log s + 4 + d(0)/\kappa}{\lambda_H \gamma^o_{\min}} \right) = \frac{2 \log s + 4 + d(0)/\kappa}{\lambda_H \gamma^o_{\min}},
$$

thus also for $\bar{\tau}$. Then under the No-false-positive property,

$$
\text{sign} \left( \gamma_S \left( \bar{\tau} \right) \right) = \text{sign} \left( \gamma'_S \left( \bar{\tau} \right) \right) = \text{sign} \left( \gamma_S^o \right) = \text{sign} \left( \gamma_S^\star \right),
$$

and
$$\text{sign}\left(\gamma'_{S^c}\left(\bar{\tau}\right)\right) = 0 = \text{sign}\left(\gamma^\star_{S^c}\right).$$

Now we prove the $\ell_2$ *consistency of* $\gamma(t)$. Under the No-false-positive property, for $0 \le t \le \bar{\tau}$,

$$\|\gamma(t) - D\beta^\star\|_2 = \|\gamma'_S(t) - \gamma^\star_S\|_2 \le \|d_{\gamma,S}(t)\|_2 + \|\gamma^o_S - \gamma^\star_S\|_2$$

$$\le d(t) + \sqrt{s}\,\|\gamma^o_S - \gamma^\star_S\|_\infty \le \frac{4\sqrt{s} + d(0)/\kappa}{\lambda_H t} + \frac{2\sigma}{\lambda_H} \cdot \frac{\Lambda_X}{\lambda_D}\sqrt{\frac{s\log m}{n}}$$

$$\le \frac{5\sqrt{s}}{\lambda_H t} + \frac{2\sigma}{\lambda_H} \cdot \frac{\Lambda_X}{\lambda_D}\sqrt{\frac{s\log m}{n}}.$$

Finally, we prove the $\ell_2$ *consistency* of $\beta(t)$. Under the No-false-positive property, for $0 \le t \le \bar{\tau}$,

$$\|\beta(t) - \beta^\star\|_2 = \|\beta'(t) - \beta^\star\|_2 \le d_\beta(t) + \|\beta^o - \beta^\star\|_2 \le d(t) + \|\beta^o - \beta^\star\|_2.$$

By Lemma 10 (especially noting (E.10)), we have

$$\|\beta^o - \beta^\star\|_2 \le \|\delta^o - \delta^\star\|_2 + \|\xi^o - \xi^\star\|_2$$

$$\le \left\|\frac{1}{\sqrt{n}}\Lambda_1^{-1}U_1^T\epsilon\right\|_2 + \left(1 + \left\|\frac{1}{\sqrt{n}}\Lambda_1^{-1}U_1^T XV\right\|_2\right) \cdot \|\delta^o - \delta^\star\|_2 \le \sqrt{r'}\left\|\frac{1}{\sqrt{n}}\Lambda_1^{-1}U_1^T\epsilon\right\|_\infty$$

$$+ \left(1 + \frac{\Lambda_X}{\lambda_1}\right)\left(\nu\left\|B^{-1}V^T X^*\left(I - U_1 U_1^T\right)\epsilon\right\|_2 + \left\|B^{-1}\Lambda U_S^T\right\|_2 \cdot \sqrt{s}\,\|\gamma^o_S - \gamma^\star_S\|_\infty\right)$$

$$\le \sqrt{r'}\left\|\frac{1}{\sqrt{n}}\Lambda_1^{-1}U_1^T\epsilon\right\|_\infty + \left(1 + \frac{\Lambda_X}{\lambda_1}\right)\left(\nu \cdot 2\sigma \cdot \frac{\Lambda_X}{\lambda_D^2} + \frac{1}{\lambda_D} \cdot \sqrt{s} \cdot \frac{2\sigma}{\lambda_H} \cdot \frac{\Lambda_X}{\lambda_D}\sqrt{\frac{\log m}{n}}\right).$$

By (A.3), with probability not less than $1 - 2/m$, we have

$$\left\|\frac{1}{\sqrt{n}}\Lambda_1^{-1}U_1^T\epsilon\right\|_\infty \le 2\sigma\left\|\frac{1}{\sqrt{n}}\Lambda_1^{-1}U_1^T\right\|_2 \sqrt{\log m} \le \frac{2\sigma}{\lambda_1}\sqrt{\frac{\log m}{n}}.$$

In this case, combining the inequalities above with $d(t) \le 5\sqrt{s}/(\lambda_H t)$, the desired result follows. $\quad\square$

*Proof of Theorem 5.* By the proof details of Theorem 4, we know that with probability not less than $1 - 6/m - 3\exp(-4n/5)$, (E.15) to (E.19) hold, meanwhile the solution path has no false-positive for $0 \le t \le \bar{\tau}$. From now, we assume that these properties are all valid.

First we prove the *sign consistency of* $\tilde{\beta}(t)$. If the $\gamma^\star_{\min}$ condition (E.2) holds, then by Theorem 4, $S(\bar{\tau}) = S$ holds, and we have

$$D_{S^c}P_{S(\bar{\tau})} = D_{S^c}\left(I - D^\dagger_{S^c}D_{S^c}\right) = 0 \implies \text{sign}\left(D_{S^c}\tilde{\beta}\left(\bar{\tau}\right)\right) = 0 = \text{sign}\left(D_{S^c}\beta^\star\right).$$

To prove $\text{sign}(D_S\tilde{\beta}(\bar{\tau})) = \text{sign}(D_S\beta^\star)$, note that

$$\left\|D_S\tilde{\beta}\left(\bar{\tau}\right) - D_S\beta^*\right\|_\infty = \left\|D_S\left(I - D^\dagger_{S^c}D_{S^c}\right)\left(\beta'\left(\bar{\tau}\right) - \beta^\star\right)\right\|_\infty$$

$$\le \left\|D_S\left(I - D^\dagger_{S^c}D_{S^c}\right)d_\beta\left(\bar{\tau}\right)\right\|_\infty + \left\|D_S\left(1 - D^\dagger_{S^c}D_{S^c}\right)\left(\beta^o - \beta^\star\right)\right\|_\infty$$

$$\le \left\|D_S\left(I - D^\dagger_{S^c}D_{S^c}\right)d_\beta\left(\bar{\tau}\right)\right\|_\infty + \|\gamma^o_S - \gamma^\star_S\|_\infty + \left\|D_S D^\dagger_{S^c}D_{S^c}\left(\beta^o - \beta^\star\right)\right\|_\infty.$$

First, by (E.20), $\kappa \ge d(0) \ge \|\gamma^o_S\|_2 \ge \gamma^o_{\min}$, and

$$\bar{\tau} \ge \frac{\log(8\Lambda_D)}{\lambda_H\gamma^o_{\min}} + \frac{2\log s + 5}{\lambda_H\gamma^o_{\min}} \ge \frac{1}{\kappa\lambda_H}\log\left(8\Lambda_D\right) + \frac{2\log s + 4 + d(0)/\kappa}{\lambda_H\gamma^o_{\min}}.$$

By (D.12), we have $d\left(\bar{\tau}\right) \le \gamma^o_{\min}/(8\Lambda_D)$, and thus

$$\left\|D_S\left(I - D^\dagger_{S^c}D_{S^c}\right)d_\beta\left(\bar{\tau}\right)\right\|_\infty$$

$$\le \|D_S\|_2 \cdot \left\|I - D^\dagger_{S^c}D_{S^c}\right\|_2 \cdot \|d_\beta\left(\bar{\tau}\right)\|_2 \le \Lambda_D \cdot d\left(\bar{\tau}\right) \le \frac{\gamma^o_{\min}}{8} \le \frac{\gamma^\star_{\min}}{4}.$$

Besides, by (E.4), we have
$$D_S D_{S^c}^\dagger D_{S^c} \left(\beta^o - \beta^\star\right) = U_S \Lambda V^T D_{S^c}^\dagger U_{S^c} \Lambda \left(\delta^o - \delta^\star\right) = U_S \Lambda V^T D_{S^c}^\dagger U_{S^c} \Lambda B_\delta \epsilon$$
with
$$\left\| U_S \Lambda V^T D_{S^c}^\dagger U_{S^c} \Lambda B_\delta \right\|_2 \le \Lambda_D \left\| D_{S^c}^\dagger \cdot U_{S^c} \Lambda V^T \right\|_2 \cdot \|B_\delta\|_2 \le \frac{\Lambda_X \Lambda_D}{\sqrt{n} \cdot \lambda_H \lambda_D^2}.$$
By (A.3), with probability not less than $1 - 2/m$,
$$\left\| D_S D_{S^c}^\dagger D_{S^c} \left(\beta^o - \beta^\star\right) \right\|_\infty < \frac{2\sigma}{\lambda_H} \cdot \frac{\Lambda_X \Lambda_D}{\lambda_D^2} \sqrt{\frac{\log m}{n}} \le \frac{\gamma_{\min}^\star}{4}.$$
Finally, we note (E.21). Then $\operatorname{sign}(D_S \tilde\beta(\bar\tau)) = \operatorname{sign}(D_S \beta^\star)$ holds, since
$$\left\| D_S \left( \tilde\beta(\bar\tau) - \beta^\star \right) \right\|_\infty < \frac{\gamma_{\min}^\star}{4} + \frac{\gamma_{\min}^\star}{2} + \frac{\gamma_{\min}^\star}{4} = (D_S \beta^\star)_{\min}.$$

Then we prove the $\ell_2$ *consistency of* $\tilde\beta(t)$. For any $0 \le t \le \bar\tau$, $S(t) \subseteq S$, which implies $D_{S^c} \tilde\beta(t) = D_{S^c} \beta^\star = 0$. Then
$$
\begin{aligned}
\left\| \tilde\beta(t) - \beta^\star \right\|_2 &\le \left\| V^T \left( \tilde\beta(t) - \beta^\star \right) \right\|_2 + \left\| V_1^T \tilde V^T \left( \tilde\beta(t) - \beta^\star \right) \right\|_2 \\
&\le \left( \left\| V^T P_{S(t)} \left(\beta'(t) - \beta^\star\right) \right\|_2 + \left\| V^T \left(I - P_{S(t)}\right) \beta^\star \right\|_2 \right) \\
&\quad + \left( \left\| V_1^T \tilde V^T P_{S(t)} \left(\beta'(t) - \beta^\star\right) \right\|_2 + \left\| V_1^T \tilde V^T \left(I - P_{S(t)}\right) \beta^\star \right\|_2 \right) \\
&\le \left\| V^T P_{S(t)} \left(\beta'(t) - \beta^\star\right) \right\|_2 + \left\| V_1^T \tilde V^T P_{S(t)} \left(\beta'(t) - \beta^\star\right) \right\|_2 + 2 \left\| D_{S(t)^c}^\dagger D_{S(t)^c \cap S} \beta^\star \right\|_2.
\end{aligned}
$$
The first and second term of the right hand side are respectively not greater than
$$
\begin{aligned}
\left\| V^T P_{S(t)} d_\beta(t) \right\|_2 + \left\| V^T P_{S(t)} \left(\beta^o - \beta^\star\right) \right\|_2 &\le \| d_\beta(t) \|_2 + \frac{1}{\lambda_D} \left\| D P_{S(t)} \left(\beta^o - \beta^\star\right) \right\|_2 \\
&\le d(t) + \frac{1}{\lambda_D} \left\| D_{S(t)} P_{S(t)} \left(\beta^o - \beta^\star\right) \right\|_2 \\
&= d(t) + \frac{1}{\lambda_D} \left\| U_{S(t)} \Lambda \left(1 - V^T D_{S(t)^c}^\dagger U_{S(t)^c} \Lambda\right) \left(\delta^o - \delta^\star\right) \right\|_2
\end{aligned}
$$
(here we use the fact that $D_{S(t)^c} P_{S(t)} = 0$), and
$$
\begin{aligned}
\left\| V_1^T \tilde V^T P_{S(t)} d_\beta(t) \right\|_2 &+ \left\| V_1^T \tilde V^T P_{S(t)} \left(\beta^o - \beta^\star\right) \right\|_2 \\
&\le \| d_\beta(t) \|_2 + \left\| \left(\xi^o - \xi^\star\right) - V_1^T \tilde V^T D_{S(t)^c}^\dagger D_{S(t)^c} \left(\beta^o - \beta^\star\right) \right\|_2 \\
&\le d(t) + \|\xi^o - \xi^\star\|_2 + \left\| V_1^T \tilde V^T D_{S(t)^c}^\dagger U_{S(t)^c} \Lambda \left(\delta^o - \delta^\star\right) \right\|_2.
\end{aligned}
$$
Noting (D.13) and (E.16), as well as applying the definition of $B_\delta$ in Lemma 10, now we only need to show that with probability not less than $1 - 2/m - 2r'/m^2$,
$$\left\| U_{S(t)} \Lambda \left(I - V^T D_{S(t)^c}^\dagger U_{S(t)^c} \Lambda\right) B_\delta \epsilon \right\|_\infty \le \frac{2\sigma}{\lambda_H} \cdot \frac{\Lambda_D \Lambda_X}{\lambda_D^2} \sqrt{\frac{\log m}{n}},$$
$$\left\| V_1^T \tilde V^T D_{S(t)^c}^\dagger U_{S(t)^c} \Lambda B_\delta \epsilon \right\|_\infty \le \frac{2\sigma}{\lambda_H} \cdot \frac{\Lambda_X}{\lambda_D^2} \sqrt{\frac{\log m}{n}},$$
which are both true, according to (A.3), as well as (E.4) which leads to
$$
\begin{aligned}
\left\| U_{S(t)} \Lambda \left(I - V^T D_{S(t)^c}^\dagger U_{S(t)^c} \Lambda\right) B_\delta \right\|_2 & \\
&\le \Lambda_D \left(1 + \left\| V^T D_{S(t)^c}^\dagger \cdot U_{S(t)^c} \Lambda V^T \right\|_2 \right) \|B_\delta\|_2 \le \frac{2\Lambda_X \Lambda_D}{\sqrt{n} \cdot \lambda_H \lambda_D^2},
\end{aligned}
$$
and
$$\left\| V_1^T \tilde V^T D_{S(t)^c}^\dagger U_{S(t)^c} \Lambda B_\delta \right\|_2 \le \left\| D_{S(t)^c} \cdot U_{S(t)^c} \Lambda V^T \right\|_2 \cdot \|B_\delta\|_2 \le \frac{\Lambda_X}{\sqrt{n} \cdot \lambda_H \lambda_D^2}.$$
$$\qquad\qquad\qquad\qquad\qquad\qquad\qquad\qquad\qquad\qquad\qquad\qquad\qquad\qquad\qquad\qquad\qquad \square$$