[Reviews · NeurIPS 2016]

Reviewer 1

Summary

This paper is interested in recovering a (non-sparse?) signal. from noisy obervations. The used loss-function ressembles the lasso but the authors clearly state the differences. An optimisation algorithm is proposed. The work provides many usual theoretical guarantees.

Qualitative Assessment

Quite like the paper, funny application (not a really important scientifically motivated question though) towards the end. More exploiting the image denoising to compare to state-of-the-art might be more useful in our community. Lots of theoretical results. Clear which should be noted, it is not always the case at all in related works! A companion empirical study would be welcome. Small questions: - l13: explain in more details the meaning/interpretation (e.g. related to practical application) of $ m $ and $ p $. Can see noise in $ \beta^{\star} $. A noisy version of $ \gamma^{\star} $? Still, it is neither observed nor measured. - l25: meaning of ADMM? - l37: 'structural' sparsity, please explain. - l41-44: rephrase please - l50-51: relationship to Champion et al. JSPI 2014? - l54 bias-free Oracle: lasso or LBI? - l66: $ (D_{1}\beta)_{p} = 0 $ ?? - l72: AUC of ROC? What about Precision vs Recall? Depends on sparsity in recovered structure... - Fig. 1 and corresponding text: include that the larger the $ \nu $, the smaller the penalisation to help with the interpretation. (e.g. l165/166) - l107: is the restriction to $ L $ a big one? In which setting? Can be assessed on data? - l109: suprised you don't need the dependence of the Hessian on $ \beta $ and $ \gamma $ since gradient with respect to these parameters are its components. Can you clarify please? - eqn (2.5) definition of $ A $ is a repeat of line 128 - l134 typo: irrEpresentable; l135: which proveS... - eqn (2.8) and ones after l152: what do these limits represent? Interpretation?? - end of 2.2 rephrase correctly to mention next section of what follows - Theorem 2 condition on $ \kappa $ (l189 and 190) are these conditions always compatible?? - l195: comment on rate - l196: what happens to the last term on the rhs of the equation? (idem in Theorem 3) - l212: how do you know $ r' $ is small? Hopefully not because of $ n $?! - l220: 'reset' ? - l226: evolves: increases?? - l227: can't see picked up random noise on Fig. 2... - Fig. 2 left: magnify!!

Confidence in this Review

2-Confident (read it all; understood it all reasonably well)


Reviewer 2

Summary

This paper proposes a novel scheme that produces a solution path for sparse linear regression. This scheme is built on top of variable splitting and the Linearized Bregman Iteration. The paper presents a theory of model selection and estimation consistency for the scheme, showing the superiority of this scheme over the generalized Lasso. The author(s) concludes the paper by evaluating this new method in two applications.

Qualitative Assessment

Model selection and estimation have been a central topic in high-dimensional linear regression, and this paper works on this important area. The new iterative method of the paper seems very interesting to me, and perhaps potentially significant. The fruitful ideas behind this method could probably be applied to other high-dimensional models, including logistic regression.

Confidence in this Review

2-Confident (read it all; understood it all reasonably well)


Reviewer 3

Summary

The Lasso penalized schemes are a class of standard methods for sparsity recovery, but they require strong incoherent conditons to ensure sparsity revovery consistency. To weaken these coherent conditons, the adaptive Lasso or some non-convexity induced methods are proposed, however, several additonal difficulties are caused among these methods, such as finding an orcale solution and the choice of adaptive parameters. From the viewpoint of algorithm, this paper presents a new iterative scheme for sparsity recovery of the linear model, based on split variables and Linearized Bregman Iteration. This method is implemented easily and specially require weaker conditions for support recovery than that in the standard Lasso. Therefore, this paper has made significant contributions in the literature of sparse models.

Qualitative Assessment

(1) It is known that some penalized least square methods with nonconvex penality (i.e. SCAD) and adaptive Lasso have been proved to recovery sparsity with high probability, under weaker conditions than the standard Lasso, so, my question is what is difference from these mentioned methods? what is novelty in the literature of sparsity induced approaches? (2) Since this paper introduces an addtional tuning parameter $nu$ compared to the existing methods, how to choose it via some specified criterion in practice?

Confidence in this Review

3-Expert (read the paper in detail, know the area, quite certain of my opinion)


Reviewer 4

Summary

The paper proposes an algorithm for the fused Lasso problem (goes with different names in different communities) by introducing an auxiliary, sparse target variable \gamma that D \beta needs to be close to. This leads to computational benefits as well as some progress towards relaxing the sufficiency condition for recovery.

Qualitative Assessment

The key idea here is not to impose sparsity directly (that would be D \beta) but on an auxiliary variable that D \beta is encouraged to be close to. This leads the authors to achieve sign consistency on the basis of a natural modification of the now standard IRR condition. Although the direct comparison with existing IRR is not obvious the authors prove that their variant is (sometimes?) weaker. A few suggestions: Right after the updates 1.4 it would be good to mention the specific loss function it minimizes (as opposed to the un-penalized, un-constrained loss 1.3). Regarding Thm 3 it would be good to clarify how \lambda_1 and \lambda_D scales in the popular random design settings.

Confidence in this Review

2-Confident (read it all; understood it all reasonably well)


Reviewer 5

Summary

This paper treats an alternative algorithm to the generalized lasso to recover signals with structural sparsity from noisy linear measurements. This algorithm has a particularly simple iterative implementation which alternates between updating the signal estimate (\beta) and the signal's representation after sparsifying transform (\gamma). A particular focus of the presentation is the interpretation of the steps of \beta and \gamma as a regularization path. It is claimed the regularization paths of Split-LBI are more accurate than those of generalized lasso in terms of area under the curve. The paper gives three results, 1) a statement that their assumptions to used guarantee consistency is weaker than the irrepresentability conditions necessary for generalized lasso as used in a prior work; 2+3) consistency results for the split-LBI algorithm and a modified version of the algorithm.

Qualitative Assessment

The paper proposes an attractively simple algorithm for solving an important problem and attempts to connect it to prior work However, too much of the paper is left to the supplementary materials and it is a bit difficult to understand the results without it. There is little discussion surrounding the main results. If more of the text of the supplementary material could be included in the main text, the paper would feel more complete. It is not completely clear why area under the curve is a useful metric to measure the improvement in regularization path over genlasso since it seems to rely on a particular setting of t (either 1/\lambda or k\alpha). More explanation on this would be helpful. There is inadequate discussion of the "revised version" of split-LBI prior to Theorem 3.

Confidence in this Review

2-Confident (read it all; understood it all reasonably well)


Reviewer 6

Summary

By going to a different regularization path, a new (convex) optimization algorithm is proposed to deal with the general Lasso type problem. Namely, new loss function is defined in (1.3) by introducing another variable $\gamma$. The consistency proof relies on several assumptions and is established solidly which appears in the appendix.

Qualitative Assessment

1. Have you ever tried ADMM (http://www.stanford.edu/~boyd/papers/admm/total_variation/total_variation_example.html) AND use some generic convex optimization solver like CVX (http://cvxr.com/cvx/)? Is your algorithm giving any competitiveness over those? 2. Does the design matrix in the examples satisfy the Assumptions in Section 2?

Confidence in this Review

1-Less confident (might not have understood significant parts)